



# Triplet State Formation of Chromophoric Dissolved Organic Matter in Atmospheric Aerosols: Characteristics and Implications

Qingcai Chen,[a*#] Zhen Mu,[a#] Li Xu,[b] Mamin Wang,[a] Jin Wang,[c] Ming Shan,[d] Xudong Yang,[d] Xingjun Fan,[e] Jianzhong Song,[f] Yuqin Wang,[a] Pengchuan Lin,[g] Lixin Zhang,[a] Zhenxing Shen,[h] Lin Du[b*]

[a] *School of Environmental Science and Engineering, Shaanxi University of Science and Technology, Xi'an 710021, China*

[b] *Environment Research Institute, Shandong University, Qingdao, 266237, China*

[c] *School of Civil Engineering, Beijing Jiaotong University, Beijing 100044, China*

[d] *Department of Building Science, School of Architecture, Tsinghua University, Beijing, 100084, China*

[e] *College of Resource and Environment, Anhui Science and Technology University, 233100 Anhui, China*

[f] *State Key Laboratory of Organic Geochemistry, Guangzhou Institute of Geochemistry, Chinese Academy of Sciences, Guangzhou 510640, China*

[g] *College of Resources and Environment, University of Chinese Academy of Sciences, 100190, Beijing, China*

[h] *Department of Environmental Science and Engineering, Xi'an Jiaotong University, Xi'an 710049, China*

*Corresponding authors:

*(Q. C.) Phone: (+86) 0029-86132765; e-mail: chenqingcai@sust.edu.cn; School of Environmental Science and Engineering, Shaanxi University of Science and Technology, Weiyang District, Xi'an, Shaanxi, 710021, China;

*(L. D.) Phone: (+86) 0532-58631980; e-mail: lindu@sdu.edu.cn; Environment Research Institute, Shandong University, Qingdao, 266237, China





**ABSTRACT:** There is chromophore dissolved organic matter (CDOM) in the
atmosphere, which may form triplet-state chromophoric dissolved organic matter
($^3$CDOM*) to further driving the formation of reactive oxygen species (ROS) under
solar illumination. $^3$CDOM* contributes significantly to aerosol photochemistry and
plays an important role in aerosol aging. We quantify the ability to form $^3$CDOM*
and drive the formation of ROS by primary, secondary and ambient aerosols. Biomass
combustion has the strongest $^3$CDOM* generation capacity and the weakest vehicle
emission capacity. Ambient aerosol has a stronger ability to generate $^3$CDOM* in
winter than in summer. Most of the triplet states generation conform to first-order
reaction, but some of them do not due to the different quenching mechanism. The
structural-activity relationship between the CDOM type and the $^3$CDOM* formation
capacity shows that the two types of CDOM identified, which similar to the
nitrogen-containing chromophores contributed 88% to the formation of $^3$CDOM*.
The estimated formation rate of $^3$CDOM* can reach ~100 µmol m$^{-3}$ h$^{-1}$ in the
atmosphere in Xi'an, China, which is approximately one hundred thousand-times the
hydroxyl radical (•OH) production. This study verified that $^3$CDOM* drives at least
30% of the singlet oxygen ($^1$O$_2$) and 31% of the •OH formed by aerosols using the
spin trapping and electron paramagnetic resonance technique.
**Keywords:** Atmospheric Chromophores; Triplet States; Structure-activity
Relationship; Excitation-emission matrices spectra (EEMs); Aerosol Photochemistry
**TOC Art:**

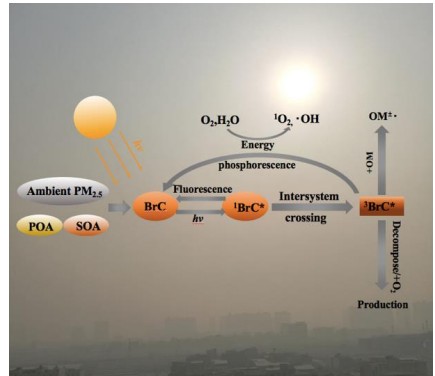



## 1 Introduction

Aerosols contain organic substances that can absorb sunlight and promote photochemical reactions and have a potentially significant impact on the global climate and atmospheric environmental quality (Borduas-Dedekind et al., 2019; Chen et al., 2016b; Chen et al., 2018; Feng et al., 2013; Rosario-Ortiz & Canonica, 2016). For example, chromophoric dissolved organic matter (CDOM) can be excited under solar illumination to form triplet state chromophoric dissolved organic matter ($^3$CDOM*) through electron transitions and intersystem crossing processes, which can drive the generation of a series of reactive oxygen species (ROS), such as hydroxyl radicals (•OH), superoxide ions ($O_2^-$) and singlet-state oxygen ($^1O_2$) (Kaur et al., 2019). Thus $^3$CDOM* has potential effects on multi-phase chemical reactions in atmospheric aerosols (Bodhipaksha et al., 2015; Grebel et al., 2011; Lin et al., 2015).

Previous studies on $^3$CDOM* are mainly about the water environment, such as sewers, terrestrial natural waters and oceans (Bodhipaksha et al., 2015; Erickson et al., 2018; Zhou et al., 2019), few studies have explored the atmospheric environment. In recent years, the CDOM in atmospheric aerosols, atmospheric fog water and rainfall has been widely found to have a strong photochemical reactivity. (Graedel & Weschler, 1981; Jacob, 1986; Kaur & Anastasio, 2018; Munger et al., 1983). For example, Corral Arroyo et al. (2018) proved that the triplet state has an effect on photochemical reaction and aerosol aging on the particle phase. Kaur et al. (2018) confirmed that CDOM in atmospheric fog water can be excited under solar illumination to form $^3$CDOM*. Smith et al. (2015) demonstrated that $^3$CDOM* contributes to secondary organic aerosol (SOA) formation under laboratory simulation conditions. $^3$CDOM* has a certain chemical reactivity, which leads to its participation in various photochemical reactions (Bluvshtein et al., 2017; Bond et al., 2013; Kloster et al., 2010; Pechony & Shindell, 2010; Sharpless et al., 2014; Zepp et al., 1985). For example, $^3$CDOM* plays an important role in the oxidation of aniline- and sulfur-containing heterocyclic pollutants (You et al., 2012). $^3$CDOM* also has the ability to convert $O_2$ molecules into ROS because the activation energy of $^3$CDOM* is higher than that of $^1O_2$ (94 kJ mol$^{-1}$) (Erickson et al., 2018; Rosario-Ortiz & Canonica, 2016). Therefore, it is important to study the formation characteristics and mechanism of $^3$CDOM* in aerosols to quantify the effect of CDOM on aerosol photochemistry.



$^3$CDOM*, as a reactive intermediate (McNeill & Canonicab, 2016; Wenk et al.,
2015), is characterized by instability, complex composition and low concentration.
Therefore, it is difficult to quantify its formation characteristics (Graber & Rudich,
2005). Studying the $^3$CDOM* quenching process by phosphorescence has become an
early analysis method. However, it is difficult to quantify the formation process and
steady-state concentration of $^3$CDOM* in this way (Chen et al., 2018; Lin et al., 2015;
Turro & Engel, 1969). Because the chemical probe method has the characteristics of
avoiding interference with the reaction system and accurate quantification (Lin et al.,
2015), the method has become a common method to study the characteristics of
$^3$CDOM* formation. At present, the main chemical probes are dimethoxyphenol,
methyl jasmonate, sorbic acid (SA) and 2,4,6-trimethylphenol (TMP) (Bodhipaksha et
al., 2015; Lin et al., 2015; Richards-Henderson et al., 2015; Rosario-Ortiz &
Canonica, 2016; Kaur & Anastasio, 2018; Moor et al., 2019; Schmitt et al., 2019),
with TMP being the most commonly used chemical probe. Lin et al. (2014) showed
the reaction ability and transformation mechanism of $^3$CDOM* formed by humic acid
using the TMP probe. Zhou et al. (2019) has reported the $^3$CDOM* energy
distribution using TMP and SA probes. Compared with other chemical probes, TMP
contains methyl-substituted groups that can be used as electron donors for $^3$CDOM*
reactions; thus, TMP has a higher reactivity (Canonica & Freiburghaus, 2001). Some
substances in the environment can inhibit the reaction between the probes and
$^3$CDOM*, but TMP is not easily affected by these substances (Canonica & Laubscher,
2008; Wenk et al., 2015). Therefore, TMP is suitable as a probe to describe the
characteristics of $^3$CDOM*.
As a precursor of $^3$CDOM*, CDOM has complex types and compositions. The
types and abilities of CDOM to form $^3$CDOM* may be different, which requires us to
analyze both the types and compositions of CDOM. Excitation-emission matrices
spectra (EEMs) are a direct method for the characterization of CDOM, allowing
identification of the chromophore types by applying parallel factor analysis
(PARAFAC), such as humic-like substances (HULIS), quinones, phenols and amino
acids (Korak et al., 2014; Ma et al., 2010; McKnight et al., 2001; Rosario-Ortiz &
Canonica, 2016; Wenk et al., 2015). Therefore, the EEM method is expected to be
used in the structure-activity relationship analysis between the CDOM types and
$^3$CDOM* formation.





The purpose of this study is to examine the formation characteristics and
mechanism of $^3$CDOM* in aerosols under solar illumination conditions. The
$^3$CDOM* formation ability, reaction kinetics and reaction mechanism by different
sources of aerosols, including primary organic aerosol (POAs), SOAs and ambient
particulate matter (PM) in Xi'an, were studied. The structure-activity relationship
between CDOM and $^3$CDOM* was also studied using the EEM-PARAFAC approach.
Finally, the environmental implications of $^3$CDOM* on aerosol photochemistry were
also revealed.

## 2 Materials and methods

### 2.1 Sample Collection.

A total of 24 ambient PM$_{2.5}$ samples, 31 POA samples and 22 SOA samples were
collected.
The ambient PM samples were collected at the Shaanxi University of Science and
Technology, Xi'an, Shaanxi (see Table S1 of SI). The PM samples were collected on
quartz fiber filters by an intelligent large-flow particle sampler (Xintuo XT-1025,
Shanghai, China) with a sampling time of 23 h 30 min and a sampling flow rate of
1000 L/min. According to the seasonal characteristics of Xi'an, 24 samples were
collected from June 24-July 5 (summer) and November 30-December 11 (winter).
The filters were stored in a refrigerator at -20 ℃ before analysis.
The sources of POA include vehicle exhaust, cooking, biomass burning and coal
combustion (see Table S2 of SI). Six vehicle exhaust samples, including medium- and
heavy-sized freight vehicles and busses (national third emission standard), were
collected on quartz filters by a customized sampler (Ma et al., 2018). Five cooking
samples, including frying (fried eggs with tomatoes and fried meat with salt) and
barbecuing (roast mutton, potatoes, squid and duck), were collected on Teflon filters
by a four-channel sampler (TH-16A, Tianhong Environmental Protection Industry,
Wuhan, China) at a flow rate of 16.7 L/min and a sampling time of 20 min. Crop
straw (artemisia, corn straw and corn straw bar-shaped compacted material, etc.),
wood (wood and grape branches) and coal (lump coal) samples were burned in farm
stoves and collected on quartz filters (tprs-001, Taipuruisi, China) by a sampler
(DustTrak 8530, TSI Inc., America). The sampling site is located in Hujiazhuang
village, Hu County, Xi'an, Shaanxi. The sampling time is 1 h, and the flow rate is 8.7


L/min. Six straw burning samples, 4 wood burning samples and 3 coal combustion
samples were collected using this method. Crop straw (wheat straw, corn straw, rice
straw, etc.) and wood (pine and Chinese fir), with 1 mL of alcohol as a combustion
promoter, were burned in a resuspension tank and collected by a sampler, thereby
replacing the filter every 10 min. Three straw burning samples and 3 wood
combustion samples were collected with this method. The crop straw and wood were
obtained in rural areas of Liuzhou, Guangxi (Wei et al., 2017; Yang et al., 2013). A
combustion sample of honeycomb-shaped coal, which is made by mixing crushed
coal with 40% clay, is collected. Honeycomb-shaped coal was ignited outside the
mixing box and then moved into the coal stove inside the box. After adding two
lumps of honeycomb-shaped coal, the mixing box was closed, and the gas pump was
opened for sampling. The coal samples were mainly obtained in Pingdingshan, Henan
(Wei et al., 2017; Yang et al., 2013). All samples are stored in a refrigerator at -20 °C
before analysis.
SOAs were obtained by oxidation of different volatile organic compounds (VOCs)
under different conditions (see Table S3 of SI) (Liu et al., 2018). The precursor VOCs
were limonene (LIM), $\alpha$-pinene (APIN), toluene (TOL) and naphthalene (NAP).
SOAs with low, moderate and high oxidation degrees are obtained by controlling the
concentrations of $O_3$, $\bullet OH$ and $NO_X$ in the reaction and the illumination conditions.
The methods of achieving low, moderate and high oxidation degrees are as follows:
(1) Low-oxidation conditions (LO): VOCs are loaded into the reaction system by
purified dry air. A high concentration of cyclohexane is used as a masking agent for
$\bullet OH$. Under pure oxygen flow conditions, $O_3$ is produced by a high voltage current,
and VOCs are oxidized by $O_3$. (2) Moderate-oxidation conditions (MO): $\bullet OH$ is
produced by photolysis of $H_2O_2$. NO is added to achieve a high $NO_X$ concentration,
and VOCs are oxidized by $NO_X$. (3) High-oxidation conditions (HO): Excess $O_3$ is an
oxidation condition. The reaction time is not shorter than 6 h to ensure the complete
reaction of VOCs. Twenty-two SOA samples were collected by a low-pressure impact
sampler (DLPI+, Dekati Ltd., Finland) with a sampling time of 1.5 h.
**2.2 Sample Extraction.**
Particle matter on filter was ultrasonically extracted for 15 min in a clean brown glass
bottle containing ultrapure water (>18.2 M$\Omega\bullet$cm, Master series, Hitech, China), and
water-soluble organic matter (WSOM) extractions were obtained through a 0.45-$\mu$m





filter (Jinteng, China). Background extractions were also obtained using blank filters
with the same method as that used for the sample extracts.

**2.3 TOC Analysis.**

A volume of 0.5 mL extracted WSOM was diluted to a concentration of 0.1-10 mg/L
with ultrapure water. The water-soluble organic carbon (WSOC) from extraction was
quantitatively analyzed by a total organic carbon (TOC) analyzer (Sievers M9,
General Electric, America) in $CO_2$ removal mode. To avoid WSOC concentration
changes caused by sample exposure to air and time prolongation, all extractions must
be quantitatively analyzed within 30 min after extraction. To avoid background
interference, background samples are also analyzed before each batch of samples is
analyzed. Before and after sample analysis, standard curves of a series of glucose
standard solutions (0, 0.05, 0.1, 0.2, 0.5, 1, 5 and 10 mg/L) were also measured.
Based on the off-line analysis mode of the TOC analyzer, each sample was
continuously analyzed 3 times, and the average value after subtracting the background
value was the final detection value. The relative standard deviation of the WSOC
content was 1.5%.

**2.4 Optical Absorption and EEM Fluorescence Spectra.**

Absorption and EEMs of the extracts were obtained using a fluorescence
spectrophotometer (Aqualog, Horiba Science, America). Detection conditions: the
excitation wavelength range is 200-600 nm and the emission wavelength range is
250-800 nm. The wavelength interval is 5 nm, and the integration time is 0.5 s. The
background samples are also analyzed under the same detection conditions and
deducted from the sample signal. The WSOC concentration in the sample was diluted
to within 10 ppm so that the absorbance at 250 nm was less than 0.5. The inner filter
effect has little influence on the results because the sample was fully diluted.
Correction of the inner filter effect for the EEMs is also performed.

**2.5 Triplet State Formation Experiments.**

Triplet state formation experiments are carried out in a customized reactor (Figure S1
of the SI). The material of the reactor is high-purity quartz. A high-purity quartz plate
is arranged at the top of the reactor. The upper edge of the reactor is embedded with a
rubber gasket, which allows sealing through clamping with the quartz plate. The
upper part of the reactor has two air vents for injecting clean air. Two holes at the
bottom of the reactor are connected with a water-cooled circulator to ensure that the



ambient temperature and humidity in the reactor are approximately 25 °C and 50%,
respectively. The reactor is placed on a magnetic agitator, and a rotation rate of 200
rpm is used to assist in stabilizing the temperature and humidity in the reactor. The
reaction was carried out in a customized quartz tank, in which 4 circular cells with a
radius of 5.6 mm and depth of 2.5 mm were set on a square quartz plate with an area
of $35{\times}35$ mm$^2$. The customized quartz tank is placed in the reactor, and the bottom
and upper edges are in contact with the water surface and air, respectively. The
illumination device is a xenon lamp with a VISREF filter (PLS-SXE 300, Perfectlight,
China; Figure S2 of the SI shows the wavelength spectrum of illumination). The
illumination intensity per unit area of the xenon lamp is approximately 1.2-1.3 times
that of sunlight at 12:00 (N34°22′35.07″, E108°58′34.58″).
Volumes of 135 µL WSOM extract (Tables S1-S3 of the SI show the WSOC
concentration information) and 25 µL TMP solution ($c_{TMP}$ = 4 mM) are mixed in the
reaction cell and reacted under simulated solar illumination. The illumination times
are 0, 5, 10, 15, 30, 45, 60 and 90 min. Samples were collected at different times.
Then, 20 µL phenol solution ($c_{phenol}$ = 4 mM), as an internal standard for quantifying
the TMP peak area of liquid chromatography, was added to the collected sample.
Compared with the previous study, the concentration of TMP used in the paper is
higher; therefore, we compared high-concentration TMP with low-concentration TMP.
The results show that under our reaction conditions, the high-concentration TMP may
have a relatively low background and a higher reaction rate constant (the results are
shown in Figure S3 of SI).
The sulfate in aerosols may produce sulfate free radicals under illumination, which
can possibly consume TMP. Simulated TMP consumption by a sulfate ion solution
was also examined in this study. Three parallel groups of background and control
experiments are compared. We studied the effect of salts on the formation of triplet
states. As shown in Figure S4, in the reaction system with or without $(NH_4)_2SO_4$, no
significant difference exists in the decay rate of TMP. To further study the effect of
salts on the formation system of triplet states, we used solid-phase extraction to
separate high-polar substance salts and low-polar HULISs (Chen, et al., 2016a). We
determined the effects of salts, HULISs, and a salt and HULIS mixture on TMP
attenuation. As shown in Figure S5, we found that when the salts were mixed with





low-polar substances, no significant effect on TMP attenuation was identified in the
low-polar reaction system.
Ultra-high-performance liquid chromatography (UPLC, Acquity UPLC H-Class,
Waters, America) was conducted to quantify the TMP concentration of the above
samples. The UPLC analysis conditions were as follows. The mobile phase consisted
of 50% acetonitrile and 50% water and had a flow rate of 0.1 mL/min; the ultraviolet
detector used a detection wavelength of 210 nm. Each batch of samples contained
internal and external standard solutions, background samples, control samples and
WSOC extraction samples. The relative standard deviation of the TMP content is

227    8.8%.

**2.6 ROS Capture Experiments.**

(1) The driving effects of the triplet state on $^1O_2$ were studied.
2,2,6,6-tetramethyl-piperidine (TEMP, $c_{TEMP}$ = 0.25 M) was used as a scavenger for
$^1O_2$, and SA ($c_{SA}$ = 4×10$^2$ μM) was added into the reaction system as a triplet state
quencher. After 60 min of illumination (the illumination device is shown in Figure S1
of the SI), the $^1O_2$ signal was detected by an electron paramagnetic resonance (EPR)
spectrometer (MS5000, Freiberg) and compared with the content of $^1O_2$ under the
condition of the presence or absence of a quencher of the triplet state. (2) The driving
effects of the triplet state on •OH were studied. 5,5-dimethyl-1-pyrroline-N-oxide
(DMPO, $c_{DMPO}$ = 0.1 M) was used as a scavenger for •OH. L-Histidine ($c_{L\text{-}Histidine}$ =
0.1 M) was used as a quencher for $^1O_2$. After 60 minutes of illumination, the content
of •OH was compared under the condition of the absence or presence of a quencher of
the triplet state. The content of •OH was also compared in the condition of the
absence or presence of a quencher of $^1O_2$.

**2.7 Data Analysis.**

Kaur and Anastasio (2018) and Richards-Henderson et al. (2015) reported that the
consumption characteristics of the probe conformed to first-order kinetics in the
reaction to form the triplet state. In this study, the value of $k_{TMP}$ is the attenuation rate
constant of TMP, which is used to calculate the yield of the triplet state in ambient
atmosphere. The first-order kinetics equation is used to fit the exponential relationship
among $k_{TMP}$, the concentration of TMP ($c_{TMP}$) and the illumination time (t):
$$c_{TMP} = ae^{tk_{TMP}} \tag{1}$$





To evaluate the contribution of different chromophore types to triplet formation and
to study the structure-activity relationship between the chromophores and triplet
formation, the PARAFAC method was used to analyze the consumption rate
constants of TMP coupled with EEM data sets. The developed model not only
identifies the types of CDOM but also identifies the relative contributions of the
different types of CDOM to the formation of $^3$CDOM*. The basic principles and
equations of the model are as follows:
$$X_{n,i,j} = \sum_{f=1}^{F} a_{n,f} \cdot \left( b_{i,f} c_{j,f} \right) + e_{n,i,j} \qquad (2)$$
where n is the number of samples for n = 1,... N; i = 1... I; j = 1,... J; k = 1,... K; $X_{n,i,j}$
are the EEM ($X_{n,\,1\ldots\,I\text{-}1,1\ldots\,J\text{-}1}$) coupled data sets of the consumption rate constants of the
TMP ($k_{TMP}$) values $X_{n,I,J}$; f is the number of factors; a is the factor load coefficient; b
and c contain factor spectrum information; and $e_{n,i,j}$ are the model residuals.
The EEM data sets coupled with $k_{TMP}$ values are analyzed through the drEEM
toolbox (http://www.models.life.ku.dk/dreem). Detailed model analysis has been
reported in previous studies by Chen et al. (2016b; 2016c). According to the EEM
characteristics of the 2- to 10-component PARAAFAC models and the trend of the
residual error, a 5-component PARAFAC model were selected.

## 267 3 Results and discussion

### 268 3.1 Reaction Kinetics of $^3$CDOM* Formation.

The kinetics characteristics of $^3$CDOM* formed by different source aerosols are
studied. Figure 1 (A) and (C) illustrate the variation of the TMP concentration under
simulated solar illumination, where (i), (ii) and (iii) are the average TMP consumption
of the POA, SOA and ambient PM samples, respectively. The results show that the
consumption of TMP in most samples conforms to the first-order kinetics equation
(Kaur & Anastasio, 2018), which indicates that the concentration of the triplet state
formed in the reaction system is constant and the formation rate of $^3$CDOM* is the
same as its quenching rate. In this case, the quenching mechanism of $^3$CDOM*
conforms to the paths (2)-(3) and (5)-(7) described in Scheme 1. In the reaction
process, $^3$CDOM* may not be consumed, but $^3$CDOM* mainly promotes energy
transfer, such as converting $O_2$ to $^1O_2$.
As shown in Figure 1C, in contrast to the above results, the consumption of TMP of
certain vehicle exhaust and biomass combustion samples do not completely conform



to first-order reaction kinetics. Specifically, the consumption of TMP within 0-20 min
of the reaction stage conforms to first-order reaction kinetics. With the reaction
proceeding, the consumption of TMP changes to zero-order reaction kinetics after 20
min. The difference in TMP consumption kinetics reflects the difference in the
quenching mechanisms of $^3$CDOM*. The results show that the reaction rate is mainly
controlled by the TMP concentration and that the $^3$CDOM* concentration remains
constant because more $^3$CDOM* is formed in the initial stage. With the reaction
proceeding, CDOM in the sample undergoes an irreversible photochemical reaction,
which results in the loss of CDOM in the reaction system so that the steady-state
concentration of $^3$CDOM* decreases. In this case, the mechanisms conform to the
paths (1) and (4) described in Scheme 1. Whether CDOM changes the molecular
properties during the process of $^3$CDOM* formation depends on its molecular
structure, as well as the type of CDOM, which is discussed in section 3.3.

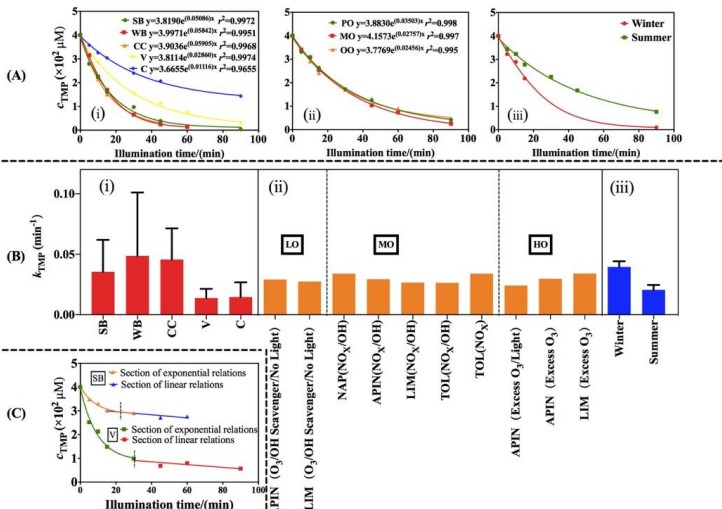


**Figure 1.** Reaction kinetics and formation rate of $^3$CDOM* of different source aerosols. (A) The average consumption of TMP of
aerosols from different sources conforms to first-order reaction kinetics. (B) Comparison of the average rates of $^3$CDOM formation of
aerosols from different sources. (C) The average consumption of TMP by aerosols does not conform to first-order kinetics. (i) POAs. (ii)
SOAs. (iii) Ambient PM. SB is straw burning, CC is coal combustion, WB is wood burning, V is vehicle exhaust and C is cooking. LO,
MO and HO indicate low oxidation, moderate oxidation and high oxidation, respectively.





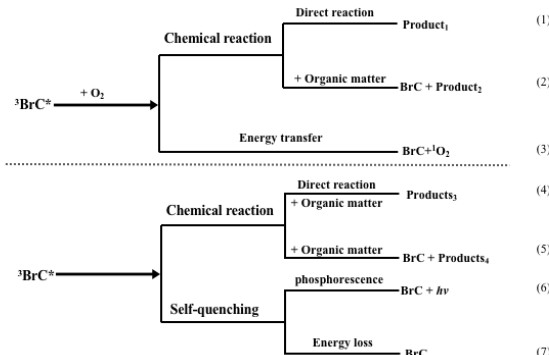


**Scheme 1.** The Quenching Mechanism of $^3$CDOM*.[1]

## 3.2 Comparison of the $^3$CDOM* Formation Ability of Aerosols Different Sources.

The $^3$CDOM* formation ability of different sources aerosols is different. As shown in Figure 1B, There is no significant difference of $k_{TMP}$ in POA, SOA and ambient PM on average, the values of $k_{TMP}$ were 0.032±0.032, 0.030±0.005 and 0.030±0.011 min$^{-1}$, respectively. However, significant differences in the $k_{TMP}$ values of POAs were found, which may be related to the large number of aromatic organic compounds produced by combustion, such as resorcinol, indole and other typical CDOM, which have a high photochemical activity (Wong et al., 2017; Glasius et al., 2006; Mcdonald et al., 2000). Straw and wood burning both belong to biomass combustion, the $k_{TMP}$ values of the straw burning samples are lower than those of the wood burning samples (0.035 min$^{-1}$). Straw burning results in large quantities of phenolic substances (Rogge et al., 1998; Schauer et al., 2001), while phenol-like chromophores do not contribute significantly to TMP consumption, as discussed in detail in section 3.3. The $k_{TMP}$ values of vehicle exhaust were the lowest (0.013 min$^{-1}$), which were similar to cooking samples. N-alkanes, carboxylic acids and alkanols, which do not produce $^3$CDOM*, are the most important markers of vehicle exhaust (Ho et al., 2009; Lee et al., 2001; Tian et al., 2009). In addition, aliphatic aldehydes and ketones account for the highest proportion of the cooking emissions in China, reaching more than 70% (Xu et al., 2017). These aliphatic compounds cannot form $^3$CDOM*, which lead to the low $k_{TMP}$ values of the vehicle exhaust and cooking samples.





The $k_{TMP}$ values of SOAs formed under different oxidation conditions are similar,
and the average value is 0.029 min⁻¹. The results indicate that the photochemical
reactivity of SOAs does not depend mainly on the precursors and oxidation degree.
The SOA samples formed by TOL under $NO_X$ oxidation conditions have a slightly
higher $k_{TMP}$ value (~1.3 times) than those formed under $NO_X$/•OH conditions, which
indicates that the CDOM formed by $NO_X$ may have a relatively higher photochemical
reactivity, but the effect is limited.
The $k_{TMP}$ value of the ambient PM in winter (0.040±0.005 min⁻¹) is approximately
2-times higher than that in summer (0.021±0.004 min⁻¹) in Xi'an, which indicate that
considerable differences exist in the types and contents of CDOM in winter and
summer. Shen et al. (2017) reported that the strong light absorption substances in
organic aerosols are a mixture of biomass burning and coal combustion emissions in
Xi'an. Increased coal combustion results in a higher content of CDOM in winter, and
CDOM from coal combustion has higher $k_{TMP}$ values, as shown in Figure 1B. On the
other hand, the most sensitive CDOM bleached or decomposed due to the high solar
illumination intensity in summer (Wong et al., 2019; Helms et al., 2008; Sharpless et
al., 2014), which results in the reduction of CDOM with photochemical reactivity.
Therefore, the ability of the ambient PM to form ³CDOM* is greater in winter. In
particular, the highest $k_{TMP}$ value was 0.046 min⁻¹ in the winter samples, which was
similar to coal combustion and wood burning samples. This result is consistent with
the fact that coal combustion is an important source of ambient PM in winter in Xi'an.
**3.3 Structure-activity Relationship Between the CDOM Types and ³CDOM***
**Formation.**
The characteristics of ³CDOM* formation depend on the chemical structure of
CDOM. Absorption spectra can provide some structural characteristics of CDOM
(Figure S6). For example, $E_2/E_3$ ($Abs_{250nm}/Abs_{365nm}$) values represent the aromaticity
and molecular weight of organic aerosols (Peuravuori & Pihlaja, 1997). Figure 2
shows the $E_2/E_3$ characteristics in the different samples. The $E_2/E_3$ values of the POA,
SOA and ambient PM samples were 6.51±3.55, 17.57±9.24 and 5.29±0.39,
respectively. As shown in Figure 3 (i)-(iii), the results show that $k_{TMP}$ has a negative
correlation with $E_2/E_3$ and a positive correlation with the mass absorption efficiency
(MAE) in all sample types. In general, a smaller $E_2/E_3$ value and a high MAE value
indicate greater aromaticity and a higher molecular weight (Duarte et al., 2005). The





results indicate that greater CDOM aromaticity corresponds to a greater $^3$CDOM*
formation ability.

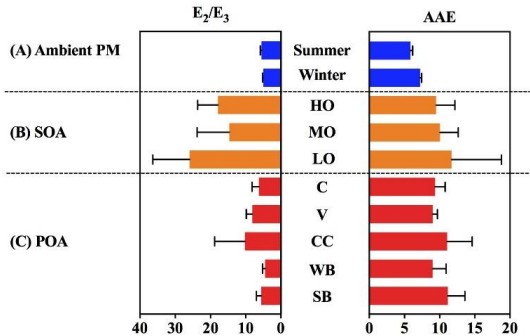


**Figure 2.** The characteristics of the AAE and $E_2/E_3$ ratio of different types of aerosols.
The absorption Angström-exponent (AAE) is also a useful parameter for
representing the chemical structure of CDOM (Wu et al., 2019). A larger AAE value
indicates a higher polarity of CDOM and degree of oxidation (Chen et al., 2016b).
The AAE values range from 5.8 to 11.7 for all samples in the wavelength range of
365-550 nm in this study (Figure 2). Additionally, the AAE values of the POA and
SOA samples were found to have a weak negative correlation with $k_{TMP}$ ($r$=0.29,
$p$<0.1235; $r$=0.19, $p$<0.4207), while the AAE values of the ambient PM samples had a
significant positive correlation with $k_{TMP}$ ($r$=0.86, $p$<0.0001, Figure 3 (d)-(f)). The
results indicate that CDOM with a higher polarity or oxidation degree in ambient PM
has a higher $^3$CDOM* formation ability, but the CDOM in POA and SOA samples
with a higher polarity or oxidation degree may have a lower $^3$CDOM* formation
ability.



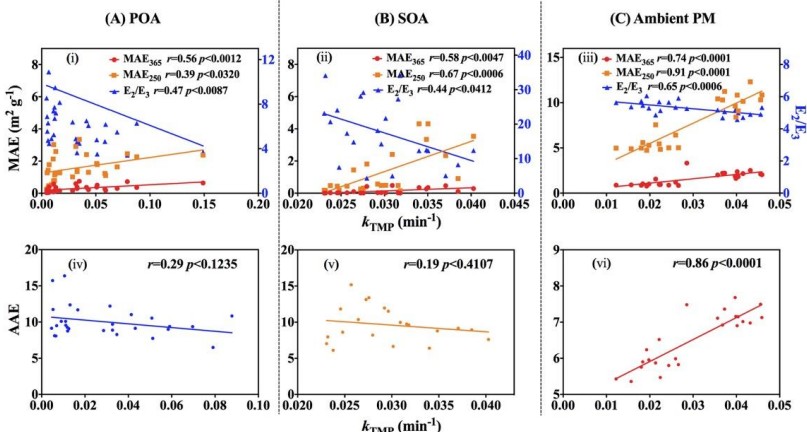

**Figure 3.** The characteristics of the correlation between $k_{TMP}$, the AAE, and the $E_2/E_3$ ratio. (i), (ii) and (iii) show the correlations between $k_{TMP}$, MAE and $E_2/E_3$. (iv), (v) and (vi) show the correlations between $k_{TMP}$ and the AAE. (A), (B) and (C) are POA, SOA and ambient PM sample results, respectively.

The $^3$CDOM* formation ability depends on the CDOM type. In this study, five types of CDOM were identified through the PARAFAC model, as shown in Figure 4C and E, and Figure 4A shows the relative contents of CDOMs in the different samples. The C1 and C3 CDOM peaks appear at Ex./Em. = 220/350 and 275/350 nm and at Ex./Em. = 230/306 and 280/306 nm, respectively, which are similar to those of tryptophan and the CDOM driven by the Maillard reaction. These peaks may be attributed to N-containing compounds (Gao & Zhang, 2018). The average content of C1 and C3 in all samples is small at only 8%. In contrast, C2 and C4 are the two CDOMs that contribute the most to fluorescence, reaching 36% and 50%, respectively, and their emission wavelengths are 352 and 430 nm, respectively, which are similar to those of less and highly oxygenated HULISs (Chen et al., 2016c; Coble, 2007; Fellman et al., 2009; Murphy et al., 2008; Wu et al., 2019). C5 contributes to the fluorescence of all samples, but only 5% on average. C5 peaks appear at Ex./Em.= 220/295 and 275/295 nm, which may be attributed to phenol-like species.



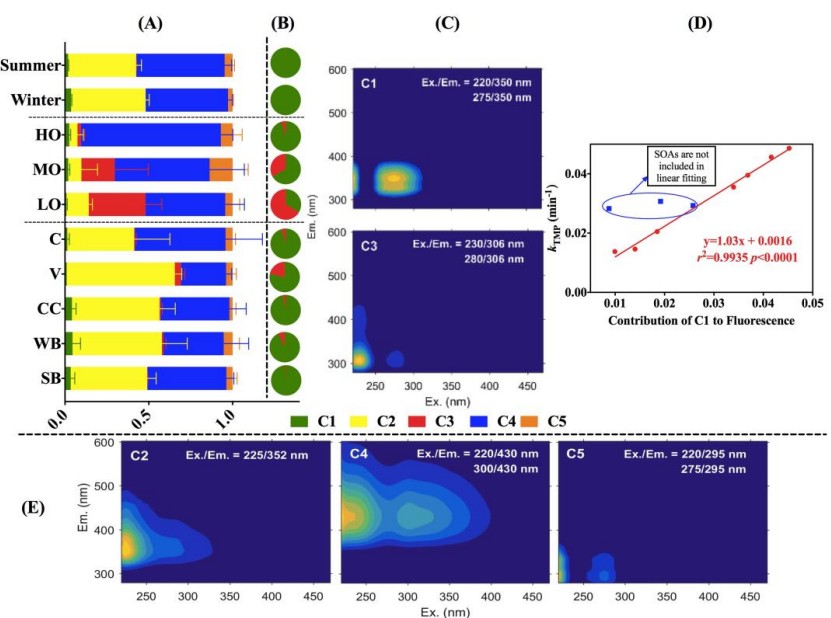

**Figure 4.** Types of CDOM and their contributions to $^3$CDOM*. (A) Different CDOM types contributing to fluorescence. (B) Difference in CDOM contributions to $^3$CDOM*. (C) EEM profiles of C1 and C3. (D) Linear relationship between the contribution of C1 to fluorescence and $k_{TMP}$. (E) EEM profiles of C2, C4 and C5.

The ability of different CDOM to form $^3$CDOM* is different. The structure-activity relationship between the CDOM type and $^3$CDOM* formation rate was established by the improved PARAFAC model in equation (2). Figure 4B illustrates the relative contributions of the different types of CDOM to the total formation rate of $^3$CDOM*. $^3$CDOM* formed by C1 and C3 contributes significantly, with average contributions of 88% and 12%, respectively. In LO-, MO- and HO-SOAs, the contributions of C1 to fluorescence are 0.9%, 1.9% and 2.5%, respectively, and the contributions to $^3$CDOM* are 33.6%, 66.8% and 95.2%, respectively. That indicates that $^3$CDOM* formation of C1 is promoted by the increase in the oxidation degree. With increasing oxidation degree, the content of C3 decreases, which indicates that C3 is oxidized and decomposed (Wong et al., 2015). C2, C4 and C5 do not contribute significantly to $^3$CDOM* formation in all samples.

The energy levels of different $^3$CDOM* is different. TMP, as a triplet-state probe used in this study, has remarkable electron transfer characteristics and is related to the energy of $^3$CDOM* (Zhou et al., 2019). Approximately 70% of high-energy $^3$CDOM*





was reportedly found in municipal wastewater using TMP as a triplet-state probe
(Zhou et al., 2019). Most $^3$CDOM* has both the ability to capture TMP by electron
transfer and to transfer energy to form $^1O_2$ (Bodhipaksha et al., 2015; McNeill &
Canonicab, 2016; Zhou et al., 2019). However, high-energy $^3$CDOM* has a greater
ability of transferring electrons (Kaur & Anastasio, 2018; Zhou et al., 2019).
According to the contribution of $^3$CDOM* to TMP consumption, the results of this
study indicated that the $^3$CDOM* formed by HULISs and phenol-like substances are
not able to transfer electrons. The quenching mechanism is mainly energy transfer,
which means that this $^3$CDOM* has more significant effect of driving ROS. However,
typical N-containing chromophores such as amino acids may include both of the
above quenching modes. Figure 3D illustrates that the contribution of C1 to
fluorescence in the POA and ambient PM samples is positively correlated with $k_{TMP}$,
indicating that C1 is the most important high-energy $^3$CDOM* precursor in aerosols.

## 4 Environmental implication

This study illustrates the $^3$CDOM* formation characteristics and mechanisms of
CDOM in aerosols. We confirm that different aerosols have the ability to form
$^3$CDOM* so that atmospheric CDOM has the potential to contribute to the
photochemical aging process of aerosols. For example, Coal combustion and wood
burning aerosols has the highest $^3$CDOM* formation ability. The results indicate that
the photochemical reactivity of aerosols from different sources is different, and that
an external mixing state of photochemical aging level exists. Based on the results of
this study, we roughly calculate the relative contribution of $^3$CDOM* to aerosol
oxidation in Xi'an, China, as shown in Table S1. The $^3$CDOM* formation rate ranges
from 52 to 194 µmol m$^{-3}$ h$^{-1}$ in this study, and the reported lifetime of $^3$CDOM* is
approximately 2-80 µs (Rosario-Ortiz & Canonica, 2016). Compared with $^3$CDOM*,
•OH is recognized as an important oxidant in aerosols. The photochemical formation
rate of •OH is approximately $(0.32\text{-}3.0) \times 10^{-3}$ µmol m$^{-3}$ h$^{-1}$, and the lifetime of •OH is
approximately 5-10 µs (Das, 2009; Faust & Allen, 1993; Lambe et al., 2007). As
stated above, the results show that the $^3$CDOM* formation rate roughly one hundred
thousand-times the •OH production, although the $^3$CDOM* reaction activity may be
thousands of times lower than that of •OH. We noted that $^3$CDOM*-involved
reactions have a potentially important contribution to the photochemical process of





atmospheric aerosols, which may be matched by that of •OH. $^3$CDOM* has been
reported to have a stronger oxidation effect on phenols and to have a significant effect
on the formation of SOAs (Smith et al., 2015).

In additional experiments, we verify the ability of $^3$CDOM* to drive ROS

formation, including $^1$O$_2$ and •OH (Manfrin et al., 2019). As shown in Figure 5, the
result shows that the signal strength of $^1$O$_2$ decreases by 30% when the $^3$CDOM* is
quenched by SA (comparing Figure 5 (c) and (d)). As shown in Figure 6, the result
shows that the signal strength of •OH decreases by 31% when the $^3$CDOM* is
quenched by SA (comparin Figure 6 (c) and (d)). We also found that the signal
strength of •OH decreases by 71% when $^1$O$_2$ is quenched by L-histidine (comparing
Figure 6 (c) and (e)). According to the above results, $^3$CDOM* drives at least 30% of
the $^1$O$_2$ and 31% of the •OH in water-soluble PM. Therefore, we speculate that
$^3$CDOM* may be a potential important driving factor for aerosol aging. This study
also shows that the aerosols from different sources have different $^3$CDOM* formation
abilities.

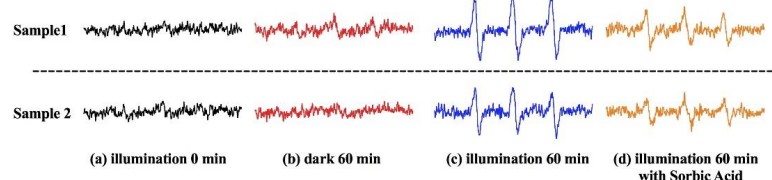


**Figure 5.** The effect of triplet states driving $^1$O$_2$. Sample 1 and Sample 2 are water extractions of two different particulate matter samples.
The particulate matter of Sample 1 and Sample 2 was collected on December 12 and 13, 2017, respectively. The concentrations of WSOC
are both 20 mg/L.

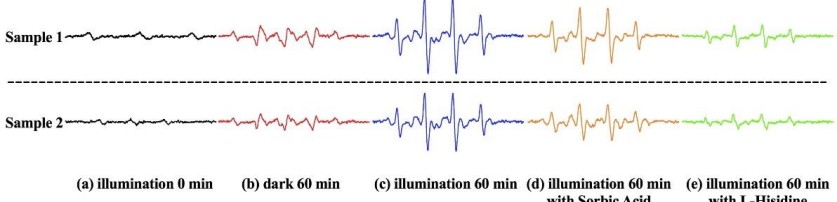


**Figure 6.** The effect of triplet states driving •OH. Sample 1 and Sample 2 are water extractions of two different particulate matter samples.
The particulate matter of Sample 1 and Sample 2 wasa collected on December 12 and 13, 2017, respectively. The concentrations of
WSOC are both 20 mg/L.



In this study, the structure-activity relationship between the CDOM type and
$^3$CDOM* formation rate was established by the EEM-PARAFAC approach. We
identify that the C1 and C3 chromophores, which may be attributed to N-containing
substances, significantly contribute to $^3$CDOM* formation, although C1 and C3
contribute little to the total fluorescence intensity. The results showed that C1 and C3
chromophores are the main precursors for the formation of $^3$CDOM* in aerosols. In
contrast, HULIS and phenol-like chromophores do not contribute significantly to
TMP attenuation. However, the above do not mean that these substances do not have
the ability to form $^3$CDOM*. In this case, as shown in Scheme 1, $^3$CDOM* through
self-quenching and energy transfer does not consume TMP, and low-energy $^3$CDOM*
cannot react with TMP.
**Data availability**
The $PM_{2.5}$ data used in this paper are from http://www.cnemc.cn (China National
Environmental Monitoring Center).
**Supporting information**
Additional details, including Tables S1−S3, Figures S1−S10, calculation of the
formation rate of $^3$CDOM* and the consumption rate of TMP due to $^3$CDOM*
formation in aerosols under solar illumination, are contained in the SI.
**Author information**
Corresponding authors:
*Q.C., phone/fax: 0086-0029-86132765; e-mail: chenqingcai@sust.edu.cn;
*L.D., phone/fax: 0086-0532-58631980; e-mail: lindu@sdu.edu.cn.
Author contributions:
#Q.C. and Z.M. contributed equally to this work.
**Acknowledgments**
This work was supported by the National Natural Science Foundation of China (grant
numbers 41877354, 41703102 and 91644214) and the Shandong Natural Science
Fund for Distinguished Young Scholars (JQ201705).





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
