# Peer review of "Triplet State Formation of Chromophoric DissolvedOrganicMatterinAtmosphericAerosols:Characteristics and Implications"

_Atmospheric Chemistry and Physics, 2019_

## Referee Comment (RC1) · Anonymous Referee #2 · 24 Feb 2020

General comments:

In this manuscript, the authors describe the formation of triplet state chromophoric dissolved organic matter (3CDOM*) from a variety of aerosol samples, including laboratory generated primary organic aerosol and secondary organic aerosol, and ambient particulate matter, under simulated sunlight. Using trimethylphenol as a probe for 3CDOM*, all aerosol samples investigated formed 3CDOM* to a greater extent than the experimental control. The rate of formation of 3CDOM* in the aerosol samples was correlated with chemical and optical properties of the sample including fluorescence and UV-visible absorbance. The contribution of the 3CDOM* to form reactive oxygen

species such as singlet molecular oxygen and hydroxyl radical was also quantified.

The experiments described in this manuscript were thoughtfully designed and executed rigorously. However, the communication of these results should be improved before publication, as indicated in my comments below. The writing style used in this manuscript makes it challenging to read at times. For example, the authors often state a conclusion before giving context or discussing the data. This requires the reader to read through the paragraph more than once to be sure of the meaning. In addition, there are grammatical issues such as in the materials and methods section which switches between past and present tense. Better clarity in the writing would greatly improve this manuscript and better communicate these interesting results. Furthermore, some of the data was presented without sufficient discussion of its meaning and its larger scientific context. For example, discussion is lacking about why the correlation of AAE and kTMP differs between laboratory generated aerosol and ambient aerosol samples; why there is a lack of linear correlation between kTMP and contribution of C1 for the SOA samples; and the meaning of the data shown in Figures 5 and 6.

Specific comments:

Lines 10 – 13: It is not clear what the two types of identified CDOM were and how that relates to nitrogen-containing chromophores.

Line 50: The topic of the reference "You et al. 2012" does not match with the text.

Lines 78 – 80: This section of text is very repetitive and uses vague terms "types and compositions" multiple times. Please clarify what you mean by CDOM "types and compositions".

Lines 140 – 147: How long are the VOCs exposed to the low-oxidation and moderate oxidation conditions? The reaction time is only mentioned for the high-oxidation condition.

Lines 203 – 204: State what you used as the "high-concentration" and "low-

concentration" for TMP.

Lines 205 – 207: The reaction rate constant should not be expected to change with TMP concentration, but the rate of the reaction will change. The calculated k value on Figure S3 shows that the rate constant does remain essentially the same between the different concentration conditions.

Figure 1: Legend for A (ii) does not match terminology in figure caption of "LO, MO, and HO". And the caption for graph C does not explain what data is presented.

Lines 305 – 307: The first two sentences of section 3.2 seem to be contradicting each other. It is stated that the 3CDOM* formation rate is different for each aerosol source, but then the data is presented to show that the average formation rate is the same for all aerosol sources.

Lines 312 – 314: The TMP rate constants should be state for both the straw and wood burning samples.

Lines 314 – 316: Phenolic compounds would be expected to be present in both the wood burning samples and straw burning samples. How do you explain the difference in kTMP for these two samples?

Lines 324 – 330: Its surprising to me that the 3CDOM* does not depend strongly on the SOA precursor. Especially since these SOA materials will have different light absorbance properties. Do you have any further insight into why the 3CDOM* formation is so similar between these samples?

Lines 353 – 355: The strength of the correlations should be state in the text. In most cases there is only a moderate correlation between MAE or E2/E3 and kTMP.

Lines 369 – 372: Why would the correlation of kTMP and AAE be opposite between POA, SOA, and ambient PM? This result should be discussed further.

Line 377: "The 3CDOM* formation ability depends on the CDOM type." The data to

support this claim is not presented until the following paragraph and it does not fit the topic of the paragraph it is in.

Line 382: The comparison of the excitation-emission peaks to tryptophan and products from the Maillard reaction should have a reference.

Lines 383 – 384: The content of C3 in the samples is quite variable and seems to be a significant fraction in the moderate-oxidation and low-oxidation SOA samples. Could this tell you more about the SOA composition under different oxidation conditions?

Figures S7 – S10 should be referenced directly in the main text.

Lines 417 – 418: It would be helpful to refer back to the reaction pathways in Scheme 1 during this discussion.

Lines 420 – 422: The lack of correlation of C1 in the SOA samples with kTMP should be discussed as well.

Line 430: It is not clear what is meant by "external mixing state of photochemical aging level".

Lines 431 – 432: The relative contribution of 3CDOM* to overall oxidation is not shown in Table S1. I could not find this data in the paper or the supporting information.

Lines 445 – 464: This section with the data on ROS production would be better suited in the Results & Discussion section of the paper.

Lines 446 – 450: The meaning of the signals and the type of sample shown in Figures 5 and 6 should be explained. As well, parts (a) and (b) should be explained in the text.

Supporting information:

Table S3: categories of oxidation should match text in main paper, with "low, moderate, and high oxidation".

Figure S3: The label for the concentration of TMP shows $4 \times 102$ $\mu$M, but this does not

[Figure]

match with the main text description that the TMP concentration is 4 mM ($4\times10^3$ $\mu$M).

Figure S4: Legend has incorrect spelling of 'ammonium'.

Technical corrections:

Line 1: "chromophore" -> "chromophoric"

Line 3: "driving" -> "drive"

Lines 6 – 8: The wording of this sentence is unclear: "Biomass combustion has the strongest 3CDOM* generation capacity and the weakest vehicle emission capacity." Is this trying to say that vehicle emissions have the weakest 3CDOM* generation?

Line 11: "structural-activity" -> "structure-activity"

Line 85: "expected" -> "well-suited"

Line 128: "is collected" -> "was collected"

Line 266: "were selected" -> "was selected"

Line 420: "Figure 3D" -> "Figure 4D"

Line 428: "has" -> "have"

Line 646: Incorrect spelling of author's name: "Canonica"

---

## Referee Comment (RC2) · Anonymous Referee #1 · 25 Feb 2020

General feedback:

The authors sampled three different types of atmospheric aerosols onto quartz filters: (1) ambient PM2.5 in Xi'an, China, (2) primary organic aerosols from biomass burning, coal combustion and vehicle exhaust, as well as (3) laboratory-generated secondary organic aerosols from smog chamber experiments using a-pinene, limonene, naphthalene and toluene in a mixture of different yet unspecified concentrations. The list of all these samples are presented in Tables S1, S2, and S3. However, the data presented in the paper seems to have regrouped these samples in different and again unspecified ways with unsupported conclusions. All raw data of the experiments are unfortu-

nately missing. Nevertheless, the TMP probe is an appropriate method for quantifying 3CDOM*, yet further controls need to be presented. For example, TMP kinetics with the blank filter and in water (two negative controls), TMP kinetics for triplicate of one sample and TMP kinetics with a known 3CDOM* as a positive control. As it stands, experiments couldn't be reproduced due to lack of details and the authors should address this issue. Furthermore, the reported TOC measurements are appropriate, yet an emphasis on the importance of concentration for the measured kinetics should be added.

A serious revamp of the manuscript writing is also recommended to present a logical sequence of events/results. For example, a discussion of the method and why the method is appropriate using logical sequencing (for example reorganizing the logic presented in lines 61-76 where the last sentence of the paragraph remains unsubstantiated and the reader is left wondering that "TMP has a higher reactivity" than what & why "TMP is not easily affected by these substances" and ). I encourage the authors to aim to be precise and concise throughout their text.

There is a missing discussion on how this study builds upon past work. For example, are the authors' results consistent (or not) with what other have observed so far? I would encourage the authors to state clearly what their hypothesis was and why they specifically chose the aerosol samples listed to support their hypothesis. Furthermore, was the starting hypothesis validated? The authors should clearly state their scientific approach.

Therefore, as it stands the paper has too little technical details, making it difficult to understand and interpret the results and thus difficult to recommend for publication in its current state.

Specific feedback:

For example, the data in Figures 1, 2, 3, 4 are incongruent with the samples described in Tables S1, S2, S3 and so clear explanations are required to follow the authors'

logic. A description of how the samples were grouped for experiments and how many experiments were conducted in total should also be added. The column for kTMP in the SI tables suggests that data was collected for each one of the samples, and so the data needs to be shown (at least in the SI). The authors should add a description of error bars/standard deviations/uncertainty of the measurement.

Acronyms: Avoid LO, MO, HO and avoid acronyms for chemical names, particularly since Atmos. Chem. Phys. does not have a word limit. Describe all acronyms in each figure caption. As it stands it is difficult to understand what is being plotted in most figures.

Title:

The title is misleading, as no "implications" of the work is mentioned.

Abstract:

Lines 4-5: the statement of 3CDOM* contributing significantly to aerosol photochemistry is a conclusion of the work, and not an introductory statement. There has yet to be studies demonstrating the impact of 3CDOM* to aging in the context of aerosol-cloud interactions.

Line 5: "the ability" of what?

Between lines 4-8: please add information on the types of aerosols investigated as well as the method used.

Line 6: be specific when mentioning primary, secondary and ambient aerosols. To some extent all these qualifies could indicate identical samples.

Line 11: The structure-activity relationship description should be made clear that it was developed in this work.

Line 16: be precise as to *how* the study verified that 3CDOM* drives 1O2. At the end: add a sentence relating to the implications of the work for aerosol photochemistry.

TOC art:

What is the meaning of the different colors/shades?

Introduction:

Lines 24-26: unsubstantiated sentence and unsupported by the chosen references. Best to use reviews on brown carbon (or even better, modeling studies) to support a claim on climate impacts, since the current references deal with laboratory studies.

Line 32: the Kaur et al. 2019 reference is only valid for the 1O2 claim in this sentence. Best to include accurate references for the other oxidants.

Lines 35-54: I encourage the authors to be more specific when referencing earlier studies. The authors should add and specify the mechanisms at play, the specific atmospheric environment (line 37), the explanation of how Corral Arroyo et al 2018 proved that the triplet state affected aerosol aging (lines 41-42), the explanation of how 3CDOM* contributes to SOA formation (line 45), the specific "certain chemical reactivity" (line 46), the specific "important role" (line 49).

Line 55: low concentrations compared to what?

Lines 58-59: consider rewriting this sentence. A method "becoming an early analysis method" appears to be an oxymoron, especially when a 1969 reference is used.

Line 61: specific the chemical probe method

Line 75: Why isn't TMP not easily affected by "these substances". Add more specific information.

Lines 84-85: specifically mention what (Korak et al., 2014; Ma et al., 2010; McKnight et al., 2001; Rosario-Ortiz & Canonica, 2016; Wenk et al., 2015) observed.

Lines 88-95: what is the study's scientific hypothesis? Why were Xi'an samples studied? How do these samples help support the hypothesis?
Material and methods:

All raw data should be included. Enough detail must be included for any scientist to be able to reproduce the data presented and be able to compare future work with this work. As the manuscript stands, these details are missing.

All details on the vehicles and buses used for exhaust PM need to be indicated. Even adding pictures of the set up could be useful for future comparisons.

Lines 103-104: a 1000 L/min flow rate for a 24 h sample appears to be very high. The PM2.5 samplers I've worked with don't typically exceed 50 L/min. The authors should comment on this high flow rate and specify the instrument used for collection.

Line 114: why is the flow rate here only 16.7 L/min? why is different than previously mentioned?

How were the quartz filters pre-conditioned before sampling?

Lines 120-133: verbs should be in the past tense. A further lack of detail, which is rather frustrating for the reader.

Line 122: specific which alcohol.

Line 136: no need for acronyms here.

Line 141: what was the concentration of cyclohexane added?

Line 142: how was ozone produced? Which high voltage? Which instrument?

Line 150: discuss why the filters were ultrasonically extracted in light of the paper, "To Sonicate or Not to Sonicate PM Filters: Reactive Oxygen Species Generation Upon Ultrasonic Irradiation"(Miljevic et al., 2014)

Lines 152-153: the filter was made of what material?

Lines 153-154: add this background extraction data to the tables in the SI.

Line 159: The Sievers M9 TOC analyzer, as far as I know is from Suez Technologies, not from General Electric...

Lines 159-161: why was sample exposure to air and time a problem. Please show this data.

Line 162: specific which background samples

Line 165: describe in detail the offline analysis method.

Line 166: why was the background subtracted? Please show the data.

Line 174: define the background samples

Lines 176-177: define the "inner filter effect" and show the data

Line 180: show dimensions of the customized reactor in Figure S1

Line 186: why are the 25 oC and 50% RH conditions chosen? Are they relevant to Xi'an?

Line 195: show the calculation (in the SI) to arrive at a factor 1.2-1.3.

Line 203: specific the "previous study" as there are no references.

Lines 208-209: support this claim with references

Lines 209-219: this paragraph is vague and lacks details. Please specific which salts were investigated and why would the authors expect a salt effect on 3CDOM*? Which literature are they building upon?

Line 159 & 221: America is a continent not a country, and should be corrected.

Lines 220-221: add information on the column used in the UPLC.

Lines 226-227: show the data for this statement in the SI.

Lines 233-234: 1O2 was quantified using EPR, how do these values compare with the

FFA method (Appiani et al., 2017)? How was the signal quantified? Which positive control was used?

Line 244: which probe was used?

Line 254: specify which types of CDOM and give examples.

Results and discussion:

I did not find any evidence to support scheme 1 in the paper. The authors should clarify how their own experiments rule out or support a particular pathway. I am rather sceptical that the measurements done in this study can differentiate between a chemical reaction and an energy transfer. How do the authors know whether the product is directly from 3CDOM* or from 1O2 + DOM? Also why did the authors chose to use the acronym BrC in this scheme when throughout the text, they use 3CDOM*? The scheme should also be made much larger and should have at least font size 12.

All figures should be separated into individual panels. For example, Figure 1 should be 3 separate figures. Why do only a few of the panels in figure 1B have error bars? Where is the error in Figure 1A? Why are there so many significant figures reported in Figure 1A; I doubt they are all meaningful. The acronyms in the figures should all be described in the caption.

Line 288: what is means by "more 3CDOM* is formed in the initial stage?

The average values reported in lines 307-308 represent which samples? What does the standard deviation represent? The authors should use IUPAC units and should report their values in seconds, rather than in minutes.

Line 308 contradicts line 307. The authors state no difference and then the authors state a significant difference. These statements need to be clarified that the seasonality of the ambient aerosols is now being discussed.

Line 312: what is the chemical difference between straw and wood burning?

Line 318: the authors claim that N-alkyl, carboxylic acids and alkanols do not produce 3CDOM*. Where is this evidence and/or this data?

Line 323: explain why aliphatic compounds cannot from 3CDOM* and why these specific compounds are attributed to the authors' result for vehicle exhaust.

Line 345: which "types" of 3CDOM* are the authors referring to? Be specific.

Figure 2: avoid all the acronyms in the middle of the figure. What do the error bars signify and why are the + values shown but not the – values?

Lines 366-367: show the data.

Figure 3: for optimal comparison, best to have all the same values for all the axes.

Figure 4 should be split into 4 or 5 individual figures. Why were SOA values not included in the fit and which mechanism explains their deviation from the fit?

Environmental implication:

Figures 5 & 6: why do the signals' noise look different in each figure?

Where and how did the authors identify C1 and C3 chromophores in the study? And how did they measure N-containing substances. None of these experiments appear in this study.

Line 472-473: this sentence is confusing. The hypothesis should be reiterated here and the results stated with the implications of the work. The authors should be comparing their work with previous work on 3CDOM* in the atmosphere in this section.

References: Appiani, E., Ossola, R., Latch, D. E., Erickson, P. R. and McNeill, K.: Aqueous singlet oxygen reaction kinetics of furfuryl alcohol: effect of temperature, pH, and salt content, Environ. Sci. Process. Impacts, 19(4), 507–516, doi:10.1039/C6EM00646A, 2017.

Miljevic, B., Hedayat, F., Stevanovic, S., Fairfull-Smith, K. E., Bottle, S. E. and Ristovski, Z. D.: To Sonicate or Not to Sonicate PM Filters: Reactive Oxygen Species Generation Upon Ultrasonic Irradiation, Aerosol Sci. Technol., 48(12), 1276–1284, doi:10.1080/02786826.2014.981330, 2014.

---

## Author Comment (AC1) · 18 Mar 2020

Thank you very much for the reviewer's suggestions for revision. According to the reviewer's suggestions, we have revised the paper. The reviewer's comments are in blue, the answers and revised text are in black.

*1. General comments*

*The authors sampled three different types of atmospheric aerosols onto quartz filters: (1) ambient PM2.5 in Xi'an, China, (2) primary organic aerosols from biomass burning, coal combustion and vehicle exhaust, as well as (3) laboratory-generated secondary organic aerosols from smog chamber experiments using a-pinene, limonene, naphthalene and toluene in a mixture of different yet unspecified concentrations. The list of all these samples are presented in Tables S1, S2, and S3. However, the data presented in the paper seems to have regrouped these samples in different and again unspecified ways with unsupported conclusions. All raw data of the experiments are unfortunately missing. Nevertheless, the TMP probe is an appropriate method for quantifying 3CDOM\*, yet further controls need to be presented. For example, TMP kinetics with the blank filter and in water (two negative controls), TMP kinetics for triplicate of one sample and TMP kinetics with a known 3CDOM\* as a positive control. As it stands, experiments couldn't be reproduced due to lack of details and the authors should address this issue. Furthermore, the reported TOC measurements are appropriate, yet an emphasis on the importance of concentration for the measured kinetics should be added. A serious revamp of the manuscript writing is also recommended to present a logical sequence of events/results. For example, a discussion of the method and why the method is appropriate using logical sequencing (for example reorganizing the logic presented in lines 61-76 where the last sentence of the paragraph remains unsubstantiated and the reader is left wondering that "TMP has a higher reactivity" than what & why "TMP is not easily affected by these substances" and). I encourage the authors to aim to be precise and concise throughout their text. There is a missing discussion on how this study builds upon past work. For example, are the authors' results consistent (or not) with what other have observed so far? I would encourage the authors to state clearly what their hypothesis was and why they specifically chose the aerosol samples listed to support their hypothesis. Furthermore, was the starting hypothesis*

*validated? The authors should clearly state their scientific approach. Therefore, as it stands the paper has too little technical details, making it difficult to understand and interpret the results and thus difficult to recommend for publication in its current state.*

Thanks for the reviewer's suggestion, this will be helpful for improving this article. We have improved the paper according to your suggestions. It mainly includes the following aspects.

1. We supplemented data background of TOC in SI.

Table S5. The background concentration of TOC analysis.

| Background ID | Concentration/ppm |
|---|---|
| 1 | 10.83 |
| 2 | 10.15 |
| 3 | 22.78 |
| 4 | 3.55 |
| 5 | 0.84 |
| 6 | 4.81 |
| 7 | 2.23 |

2. We supplemented data of triplet state background and parallel experiments in SI.

Table S6. Background of of triplet state formation.

| Parallel experiments | $k_{TMP}/min^{-1}$ |
|---|---|
| 1 | 0.018 |
| 2 | 0.019 |
| 3 | 0.017 |

Table S7. Analysis results of parallel experiments of triplet state formation.

| Parallel experiments | $k_{TMP}/min^{-1}$ |
|---|---|
| 1 | 0.041 |
| 2 | 0.042 |
| 3 | 0.043 |

3. We have corrected words in lines 61-76.

We have corrected "*Because the chemical probe method has the characteristics of avoiding interference with the reaction system and accurate quantification (Lin et al., 2015) ......Some substances in the environment can inhibit the reaction between the probes and 3CDOM\*, but TMP is not easily affected by these substances (Canonica & Laubscher, 2008; Wenk et al., 2015)*" to "*Chemical probe method has become a common method to study the characteristics of $^{3}CDOM$\* formation (Lin et al., 2015) and the main chemical probes are dimethoxyphenol, ......which is unaffected by CDOM inhibition effects (Canonica & Freiburghaus, 2001; Canonica & Laubscher, 2008; Wenk et al., 2015)*".

*2. Specific comments*

*Title:*

*1. The title is misleading, as no "implications" of the work is mentioned.*

We have corrected "*implications*" to "*effect*".

*Abstract:*

*1. Lines 4-5: the statement of 3CDOM\* contributing significantly to aerosol photochemistry is a conclusion of the work, and not an introductory statement. There has yet to be studies demonstrating the impact of 3CDOM\* to aging in the context of aerosol-cloud interactions.*

According to the comment, we have changed "$^3CDOM^*$ *contributes*" to "*Thus,* $^3CDOM^*$ *may contributes*".

In this study, we have proved that CDOM in aerosols can form $^3CDOM^*$, which drives the generation of reactive oxygen species. And this work demonstrated hydroxyl and singlet oxygen are important driving factors of aerosol aging. Therefore, we state that the $^3CDOM^*$ can affect the aerosols aging by producing reactive oxygen species.

*2. Line 5: "the ability" of what?*

We have corrected "*We quantify the ability to form* $^3CDOM^*$ *and drive the formation of ROS by primary, secondary and ambient aerosols*" to "*We quantify the ability of CDOM of primary, secondary and ambient aerosols forming* $^3CDOM^*$ *and* $^3CDOM^*$ *driving the formation of ROS*".

*3. Between lines 4-8: please add information on the types of aerosols investigated as well as the method used.*

We've already introduced the collection method and detailed information of aerosol samples used in the study in section 2.1.

*4. Line 6: be specific when mentioning primary, secondary and ambient aerosols. To some extent all these qualifies could indicate identical samples.*

The detail of POA, SOA and Ambient PM is shown in Table S1-S3 of SI.

*5. Line 11: The structure-activity relationship description should be made clear that it was developed in this work.*

We have added "*The structure-activity relationship reveals the contribution of CDOM to $^3$CDOM\* formation*" in the improved paper.

*6. Line 16: be precise as to \*how\* the study verified that 3CDOM\* drives 1O2. At the end: add a sentence relating to the implications of the work for aerosol photochemistry.*

By comparing the amount of $^1O_2$ and ·OH in the reaction system with or without triplet state, we prove the contribution of triplet state to reactive oxygen species.

We have added "*which reflects the significant contribution of triplet state to aerosol aging*" in the improved paper.

*TOC art:*

*1. What is the meaning of the different colors/shades?*

We has modified the TOC. As shown in TOC, the ellipse represents three sources of aerosols and the square represents different states of brown carbon.

[Figure]

*Introduction:*

*1. Lines 24-26: unsubstantiated sentence and unsupported by the chosen references. Best to use reviews on brown carbon (or even better, modeling studies) to support a claim on climate impacts, since the current references deal with laboratory studies.*

We have corrected "*Aerosols contain organic substances that can absorb sunlight and promote photochemical reactions and have a potentially significant impact on the global climate and atmospheric environmental quality*" to "*Aerosols contain organic substances that can absorb sunlight and promote photochemical reactions and have a potentially significant impact on atmospheric photochemical reaction process and atmospheric quality*".

*2. Line 32: the Kaur et al. 2019 reference is only valid for the 1O2 claim in this sentence. Best to include accurate references for the other oxidants.*

We have added references in the improved paper.

➢ Fujii, M.; Rose, A. L.; Waite, T. D. and Omura, T.: Oxygen and superoxide-mediated redox kinetics of iron complexed by humic substances in coastal seawater. Environ. Sci. Technol., 44, 9337−9342, https://doi.org/10.1021/es102583c, 2010.

➢ Li, R., Zhao, C., Yao, B., Li, D., Yan, S. W., O'Shea, K. E. and Song, W. H.: Photochemical Transformation of Aminoglycoside Antibiotics in Simulated Natural Waters, Environ. Sci. Technol., 50, 2921-2930, https://doi.org/10.1021/acs.est.5b05234, 2016.

➢ Sun, L. N.; Qian, J. G.; Blough, N. V. and Mopper, K.: Insights into the photoproduction sites of hydroxyl radicals by dissolved organic matter in natural waters, Environ. Sci. Technol. Lett., 2, 352−356, https://doi.org/10.1021/acs.estlett.5b00294, 2015.

➢ Zhang, D. N.; Yan, S. W. and Song, W. H.: Photochemically induced formation of reactive oxygen species (ROS) from effluent organic matter, Environ. Sci. Technol., 48, 12645−12653, https://doi.org/10.1021/es5028663, 2014.

*3. Lines 35-54: I encourage the authors to be more specific when referencing earlier studies. The authors should add and specify the mechanisms at play, the specific atmospheric environment (line 37), the explanation of how Corral Arroyo et al 2018 proved that the triplet state affected aerosol aging (lines 41-42), the explanation of how 3CDOM\* contributes to SOA formation (line 45), the specific "certain chemical reactivity" (line 46), the specific "important role" (line 49).*

According to the comment, we have corrected those points in the improved paper.

We have corrected "*such as sewers, terrestrial natural waters and oceans (Bodhipaksha et al., 2015; Erickson et al., 2018; Zhou et al., 2019)*" to "*CDOM in sewers (Bodhipaksha*

*et al., 2015) and river (Erickson et al., 2018; Zhou et al., 2019) have the ability of formation $^3CDOM*$*".

We have corrected "*Corral Arroyo et al. (2018) proved that the triplet state has an effect on photochemical reaction and aerosol aging on the particle phase*" to "*Corral Arroyo et al. (2018) proved that Atmospheric particles contain BrC, which is the triplet state forming organic compounds that can sustain catalytic radical reactions and thus contribute to oxidative aerosol aging*".

We have corrected "*$^3CDOM*$ has a certain chemical reactivity, which leads to its participation in various photochemical reactions*" to "*$^3CDOM*$ participation in various photochemical reactions have been revealed*".

*4. Line 55: low concentrations compared to what?*

We have deleted "*low concentrations*".

*5. Lines 58-59: consider rewriting this sentence. A method "becoming an early analysis method" appears to be an oxymoron, especially when a 1969 reference is used.*

We have corrected "*Studying the $^3CDOM*$ quenching process by phosphorescence has become an early analysis method*" to "*The quenching process of $^3CDOM*$ could be studied by phosphorescence*".

*6. Line 61: specific the chemical probe method*

In this study, the chemical probe method is as follows: specific chemical substance reacts with triplet state, and the formation characteristics of triplet state are indirectly studied by quantitative analysis of the specific chemical substance.

*7. Line 75: Why isn't TMP not easily affected by "these substances". Add more specific information. Dissolved organic matter (DOM) is both a promoter and an inhibitor of triplet-induced organic contaminant oxidation.*

We have corrected "*Some substances in the environment can inhibit the reaction between the probes and $^3CDOM*$, but TMP is not easily affected by these substances*" to "*In addition, CDOM electronic absorption in the visible range is largely due to donor-*

*acceptor complexes between electron-rich aromatic donors and carbonyl-containing acceptors. The inhibitory effect decreased with the increasing extent of CDOM pre-oxidation, and it was correlated to the loss of phenolic antioxidant moieties, as quantified electrochemically, and to the loss of DOM ultraviolet absorbance. The triplet photosensitizing ability of pre-oxidized DOM was determined using the conversion of the probe compound TMP, which is unaffected by CDOM inhibition effects*".

> ➢ Wenk, J., Aeschbacher, M., Sander, M., Gunten, U. V., and Canonica S.: Photosensitizing and Inhibitory Effects of Ozonated Dissolved Organic Matter on Triplet-Induced Contaminant Transformation, Environ. Sci. Tech., 49, 8541-8549, https://doi.org/10.1021/acs.est.5b02221, 2015.

*8. Lines 84-85: specifically mention what (Korak et al., 2014; Ma et al., 2010; McKnight et al., 2001; Rosario-Ortiz & Canonica, 2016; Wenk et al., 2015) observed.*

We have corrected "*such as humic-like substances (HULIS), quinones, phenols and amino acids*" to "*For example, humic-like substances (HULIS), quinones, phenols and amino acids could be identified by fluorescence spectral characteristics*".

*9. Lines 88-95: what is the study's scientific hypothesis? Why were Xi'an samples studied? How do these samples help support the hypothesis?*

Scientific hypothesis: We think that there are a lot of chromophore substances in aerosols, which not only affect the solar radiation, but also affect the atmospheric photochemical reaction process. Therefore, we insist that chromophores in aerosols have the ability to form trilinear States, and further affect the formation of reactive oxygen species and the aging of aerosols.

As the largest central city in Northwest China, Xi'an city has serious air pollution and high annual average concentration of carbonaceous aerosols. The higher carbonaceous components contain more chromophores, so we chose the aerosol in Xi'an as the research object.

> ➢ Mu, Z., Chen, Q. C., Wang, Y. Q., Shen, Z. X., Hua, X. Y., Zhang, Z. M. et al.: Characteristics of carbonaceous aerosol pollution in $PM_{2.5}$ in Xi'an, 40, 1529-1536, Environ. Sci., http://doi.org/10.13227/j.hjkx.201807135,2019.

We think that the types, contents and the ability to form triplet states of chromospheres are different in different sources of aerosols. Therefore, different sources of aerosols have been studied to comprehensively explain the characteristics of the formation of triplet states in different sources of aerosols.

*Material and methods:*

*1. Lines 103-104: a 1000 L/min flow rate for a 24 h sample appears to be very high. The PM2.5 samplers I've worked with don't typically exceed 50 L/min. The authors should comment on this high flow rate and specify the instrument used for collection.*

In section 2.1, we illustrate the details of particulate matter collection by an intelligent large-flow particle sampler (Xintuo XT-1025, Shanghai, China). A large sampling flow is accompanied by a large area of quartz fiber filter, which is 348.8 cm$^2$.

*2. Line 114: why is the flow rate here only 16.7 L/min? why is different than previously mentioned?*

Compared to ambient PM, the concentration of POA is very high, and the filter area we use is 43.01 cm$^2$. Therefore, according to aerosol concentration, sampling conditions, filter area, small sampling flow rate of POA was selected.

*3. How were the quartz filters pre-conditioned before sampling?*

Before sampling, quartz filter is baked in muffle furnace at 450 ℃ for 4.5. Quartz filter is put it into a clean tin foil bag after baking.

*4. Lines 120-133: verbs should be in the past tense. A further lack of detail, which is rather frustrating for the reader.*

We have corrected it in the improved paper.

*5. Line 122: specific which alcohol.*

We have corrected "*alcohol*" to "*ethanol*".

*6. Line 136: no need for acronyms here.*

We have deleted acronyms of limonene, α-pinene, toluene and naphthalene.

*7. Line 141: what was the concentration of cyclohexane added?*

We have corrected "*cyclohexane*" to "*cyclohexene*". The concentration of cyclohexene is $2.9 \times 10^{13}$ molecule・$cm^{-3}$.

- ➤ Liu, S. J., Jiang, X. T., Tsona, N. T., Lv, C., and Du, L.: Effects of NOx, SO2 and RH on the SOA Formation from Cyclohexene Photooxidation, Chemosphere, 216, 794-804, https://doi.org/10.1016/j.chemosphere.2018.10.180, 2018.

*8. Line 142: how was ozone produced? Which high voltage? Which instrument?*

We have corrected "*Under pure oxygen flow conditions, $O_3$ is produced by a high voltage current, and VOCs are oxidized by $O_3$*" to "*and VOCs were oxidized by $O_3$*".

- ➤ Liu, S. J., Jiang, X. T., Tsona, N. T., Lv, C., and Du, L.: Effects of NOx, SO2 and RH on the SOA Formation from Cyclohexene Photooxidation, Chemosphere, 216, 794-804, https://doi.org/10.1016/j.chemosphere.2018.10.180, 2018.

*9. Line 150: discuss why the filters were ultrasonically extracted in light of the paper, "To Sonicate or Not to Sonicate PM Filters: Reactive Oxygen Species Generation Upon Ultrasonic Irradiation" (Miljevic et al., 2014)*

Our previous research has proved that the efficiency of ultrasonic extraction for water-soluble brown carbon is high (the extraction efficiency of fluorescent substances is 77%, and the extraction efficiency of light-absorbing substances is 89.3%).

- ➤ Chen, Q. C., Mu, Z., Song, W. H., Wang, Y. Q., Yang, Z. H., Zhang, L. X. & Zhang, Y. L.: Size-resolved characterization of the chromophores in atmospheric particulate matter from a typical coal-burning city in China, 124, https://doi.org/10.1029/2019JD031149, 2019.

Ultrasound may cause the production of reactive oxygen species. However, for example, the lifetime of hydroxyl radical is only 5-10 μs. During the extraction process, we did not add any reactive oxygen species capture agent. Therefore, we do not think that the reactive

oxygen species generated by ultrasonic extraction will affect the study of the triplet driving active oxygen.

**10. Lines 152-153: the filter was made of what material?**

We have corrected "*a 0.45-µm filter*" to "*a 0.45-µm quartz filter*".

**11. Lines 153-154: add this background extraction data to the tables in the SI.**

We have added it in SI.

Table S5. The background concentration of TOC analysis.

| Background ID | Concentration/mg/L |
|:---:|:---:|
| 1 | 10.83 |
| 2 | 10.15 |
| 3 | 22.78 |
| 4 | 3.55 |
| 5 | 0.84 |
| 6 | 4.81 |
| 7 | 2.23 |

**12. Line 159: The Sievers M9 TOC analyzer, as far as I know is from Suez Technologies, not from General Electric.**

When we bought the instrument, it was still owned by General Electric Company. We has referenced the mark marked on the instrument.

**13. Lines 159-161: why was sample exposure to air and time a problem. Please show this data.**

According to our previous analysis methods and research experience, the detection value of TOC of WSOC extraction sample will increase if it is placed or exposed to air for a long time.

**14. Line 162: specific which background samples.**

Each time a batch of TOC analysis is performed, the background sample is analyzed in the same way, Table S5 of SI.

*15. Line 165: describe in detail the offline analysis method.*

The off-line analysis mode is the internal program of the TOC analyzer. Each sample analysis requires a fresh start. Detailed is shown in section 2.3.

*16. Line 166: why was the background subtracted? Please show the data.*

Background is subtracted for more accurate concentration information of WSOC. Background detail is shown in Table S5 of SI.

*17. Line 174: define the background samples*

The background samples are the sample without air sampling process, and the rest process is the same as the samples.

*18. Lines 176-177: define the "inner filter effect" and show the data*

Inner filter effect: When the concentration of extraction is high, the fluorescence will be weakened because of the absorption of excitation or emission light by Light absorbing substance. During the experiment, we only reduce the concentration of the extraction liquid to avoid the internal filtration effect. It is a fundamental principle in EEM analysis. When the absorbance is 0.5 at most, the internal rate effect coefficient is about $3.16 \approx 10^{(Abs\_Ex+Abs\_Em)/2} = 10^{(0.5+0.5)/2}$ at most. The instrument (Aqualog, Horiba Science) matching data collection software also has the automatic correction function.

> ➢ Murphy, K. R.; Stedmon, C. A.; Graeber, D.; Bro, R., Fluorescence spectroscopy and multi-way techniques. PARAFAC. Anal. Methods 2013, 5 (23), 6557-6566.

*19. Line 180: show dimensions of the customized reactor in Figure S1.*

We have corrected it in Figure S1. "*The inside diameter of the reaction device is about 11 cm and the height of the reaction device is about 18 cm.*"

*20. Line 186: why are the 25 °C and 50% RH conditions chosen? Are they relevant to Xi'an?*

Our custom quartz plate is open, 50% humidity and temperature of 25 °C can not only ensure that the sample solution does not volatilize largely during the illumination process, but also provide a suitable reaction environment and reaction temperature.

*21. Line 195: show the calculation (in the SI) to arrive at a factor 1.2-1.3.*

We calculated the intensity of the illumination by *Tropospheric Visible Ultra-Violet (TUV) model web page.* Input parameters for the TUV model were: Longitude: E108°58′34.58″, Latitude: N34°22′35.07″, measurement altitude: 0.02 km, surface albedo: 0.1, aerosol optical depth: 0.235, cloud optical depth: 0.00.

*22. Line 203: specific the "previous study" as there are no references.*

We have added references.

➤ Bodhipaksha, L. C., Sharpless, C. M., Chin, Y. P., Sander, M., Langston, W. K., and MacKay, A. A.: Triplet Photochemistry of Effluent and Natural Organic Matter in Whole Water and Isolates from Effluent-Receiving Rivers, Environ. Sci. Technol., 49, 3453-3463, http://doi.org/10.1021/es505081w, 2015.

➤ Zhou, H. X., Yan, S. W., Lian, L. S., and Song. W. H.: Triplet-State Photochemistry of Dissolved Organic Matter: Triplet- State Energy Distribution and Surface Electric Charge Conditions, Environ. Sci. Tech., 53, 2482-2490, https://doi.org/10.1021/acs.est.8b06574, 2019.

*23. Lines 208-209: support this claim with references*

We have added a reference.

➤ Dogliotti, L & Hayon, E.:Flash photolysis of per[oxydi]sulfate ions in aqueous solutions. The sulfate and ozonide radical anions, J. Phys. Chem., 71, 2511-2516, https://doi-org/10.1021/j100867a019, 1967.

*24. Lines 209-219: this paragraph is vague and lacks details. Please specific which salts were investigated and why would the authors expect a salt effect on 3CDOM*? Which literature are they building upon?*

In this study, $NH_4^+$, $SO_4^{2-}$, $NO_3^-$ and $Ca^{2+}$ were the main components of water-soluble ions in $PM_{2.5}$ in Xi'an (Li et al. manuscript, in review), And sulfate ion can generate sulfate free radical. Sulfate was selected in order to demonstrate that free radicals formed by ions do not consume TMP.

➢ Dogliotti, L & Hayon, E.:Flash photolysis of per[oxydi]sulfate ions in aqueous solutions. The sulfate and ozonide radical anions, J. Phys. Chem., 71, 2511-2516, https://doi-org/10.1021/j100867a019, 1967.

*25. Line 159 & 221: America is a continent not a country, and should be corrected.*

We have corrected "*America*" to "*the US*".

*26. Lines 220-221: add information on the column used in the UPLC.*

We have added "*Column type is C18 and length is 12 cm*" in the improved paper.

*27. Lines 226-227: show the data for this statement in the SI.*

We have added the data in SI.

Table S7. Analysis results of parallel experiments of triplet state formation.

| Parallel experiments | $k_{TMP}$/min$^{-1}$ |
|---|---|
| 1 | 0.041 |
| 2 | 0.042 |
| 3 | 0.043 |

*28. Lines 233-234: 1O2 was quantified using EPR, how do these values compare with the FFA method (Appiani et al., 2017)? How was the signal quantified? Which positive control was used?*

It is a simple method to quantify the production of active oxygen by chemical probe FFA. The main quantitative method is the calculation of quantum yield. However, the EPR method is a direct method, and the main measurement method is the number of spin electrons. Our main purpose is to prove that the triplet state has a significant driving effect on reactive oxygen species. We have not compared the two methods and this is also the direction of our future research.

*29. Line 244: which probe was used?*

We have corrected "*the probe*" to "*TMP*".

*30. Line 254: specify which types of CDOM and give examples.*

We have explained in detail in section 3.3. Such as tryptophan may have more significant ability of driving triplet state.

*Results and discussion:*

*I did not find any evidence to support scheme 1 in the paper. The authors should clarify how their own experiments rule out or support a particular pathway. I am rather sceptical that the measurements done in this study can differentiate between a chemical reaction and an energy transfer. How do the authors know whether the product is directly from 3CDOM\* or from 1O2 + DOM? Also why did the authors chose to use the acronym BrC in this scheme when throughout the text, they use 3CDOM\*? The scheme should also be made much larger and should have at least font size 12. All figures should be separated into individual panels. For example, Figure 1 should be 3 separate figures. Why do only a few of the panels in figure 1B have error bars? Where is the error in Figure 1A? Why are there so many significant figures reported in Figure 1A; I doubt they are all meaningful. The acronyms in the figures should all be described in the caption.*

The reaction mechanism of scheme 1 has been proven in previous study, which is mainly used to speculate on the reaction mechanism.

➢ Rosario-Ortiz, F. L., and Canonica, S.: Probe Compounds to Assess the Photochemical Activity of Dissolved Organic Matter, Environ. Sci. Tech., 50, 12532-12547, https://doi.org/10.1021/acs.est.6b02776, 2016.

We have corrected "*BrC*" to "*CDOM*".

According to reviewer suggestions, we also corrected the Figure 1.

[Figure]

*1. Line 288: what is means by "more 3CDOM* is formed in the initial stage?*

We consider that there exists following-mentioned reaction process. In the early stage of illumination, the triplet state formed by CDOM is the main reaction. The degree of triplet reaction with TMP is relatively weak, so it has a certain amount of triplet accumulation. Therefore, we believe that "*more $^3CDOM*$ is formed in the initial stage*".

$$CDOM \xrightarrow{+ h\nu} {}^3CDOM* \xrightarrow{+ TMP} Product$$

*2. The average values reported in lines 307-308 represent which samples? What does the standard deviation represent? The authors should use IUPAC units and should report their values in seconds, rather than in minutes.*

We have corrected "*the values of $k_{TMP}$ were 0.032±0.032, 0.030±0.005 and 0.030±0.011 min$^{-1}$, respectively*" to "*which were 0.032±0.032, 0.030±0.005 and 0.030±0.011 min$^{-1}$*".

The lighting stage and the curve fitting of TMP both are in minutes.

*3. Line 308 contradicts line 307. The authors state no difference and then the authors state a significant difference. These statements need to be clarified that the seasonality of the ambient aerosols is now being discussed.*

The average value of POA, SOA and Ambient PM from different sources is similar. For example, $k_{TMP}$ of POA, SOA and Ambient PM were 0.032±0.032, 0.030±0.005 and

0.030±0.011 min-1, respectively. Coal burning, straw burning, and motor vehicles belong to POA, their values are significantly different.

*4. Line 312: what is the chemical difference between straw and wood burning?*

The straw we mentioned here is mainly agricultural crops. At the same time, there may be more phenols in the combustion products of straw, but wood does not have these characteristics.

*5. Line 318: the authors claim that N-alkyl, carboxylic acids and alkanols do not produce 3CDOM\*. Where is this evidence and/or this data?*

Because of the complex nature of the aerosol composition, current studies cannot accurately determine the precursors of triplet formation, but aromatic ketones, aldehydes and quinones may be typical of triplet state precursor. We have corrected to "*which do not produce $^3CDOM*$*" to "*which may be not typical $^3CDOM*$ precursor*".

> ➤ Rosario-Ortiz, F. L., and Canonica, S.: Probe Compounds to Assess the Photochemical Activity of Dissolved Organic Matter, Environ. Sci. Tech., 50, 12532-12547, https://doi.org/10.1021/acs.est.6b02776, 2016.
> ➤ Ma, J. H., Del Vecchio, R., Golanoski, K. S., Boyle, E. S., and Blough, N. V.: Optical Properties of Humic Substances and CDOM: Effects of Borohydride Reduction, Environ. Sci. Tech., 44, 5395–5402, https://doi.org/10.1021/es100880q, 2010.
> ➤ Wenk, J., Aeschbacher, M., Sander, M., Gunten, U. V., and Canonica S.: Photosensitizing and Inhibitory Effects of Ozonated Dissolved Organic Matter on Triplet-Induced Contaminant Transformation, Environ. Sci. Tech., 49, 8541-8549, https://doi.org/10.1021/acs.est.5b02221, 2015.

*6. Line 323: explain why aliphatic compounds cannot from 3CDOM\* and why these specific compounds are attributed to the authors' result for vehicle exhaust.*

Similar to the previous question, there is currently no precise judgment on the precursors in the triplet state. We have made a reasonable guess based on the study results. We have corrected "*These aliphatic compounds cannot form $^3CDOM*$, which lead to the low $k_{TMP}$*

values of the vehicle exhaust and cooking samples" to "*These substances do not contribute significantly to the triplet state*".

*7. Line 345: which "types" of 3CDOM\* are the authors referring to? Be specific.*

What we mentioned in the paper are the types of CDOM. From the experimental results, we insist that the ability of different CDOM to form triplets is different. We have explained in detail in section 3.3.

*8. Figure 2: avoid all the acronyms in the middle of the figure. What do the error bars signify and why are the + values shown but not the–values?*

We have corrected Figure 2.

[Figure]

*Figure 2. The characteristics of the AAE and E$_2$/E$_3$ ratio of different types of aerosols.*

The data are shown in Figure 3.

[Figure]

*Figure 3. The characteristics of the correlation between $k_{TMP}$, the AAE, and the $E_2/E_3$ ratio. (i), (ii) and (iii) show the correlations between $k_{TMP}$, MAE and $E_2/E_3$. (iv), (v) and (vi) show the correlations between $k_{TMP}$ and the AAE. (A), (B) and (C) are POA, SOA and ambient PM sample results, respectively.*

*10. Figure 3: for optimal comparison, best to have all the same values for all the axes.*

We have corrected Figure 3. As shown in the figure above.

*11. Figure 4 should be split into 4 or 5 individual figures. Why were SOA values not included in the fit and which mechanism explains their deviation from the fit?*

Figure 4 have been split into Figure 4 and 5 in the paper.

[Figure]

*Figure 4. 5 types of CDOM. (A) CDOM contributing to triplet formation. (B) CDOM that do not contribute to the triplet state.*

[Figure]

*Figure 5. Types of CDOM and their contributions to $^3CDOM^*$. (A) Different CDOM types contributing to fluorescence. (B) Difference in CDOM contributions to $^3CDOM^*$. (C) Linear relationship between the contribution of C1 to fluorescence and $k_{TMP}$.*

We found that the contribution of C1 to fluorescence did not significantly affect the $k_{TMP}$. As shown in the Figure 5, SOA do not include in the linear fitting.

*Environmental implication:*

*1. Figures 5 & 6: why do the signals' noise look different in each figure?*

When the reactive oxygen species were studied by EPRs, the same curves of the two groups could not be detected. According to the characteristic peaks, the generation characteristics of different reactive oxygen species are judged.

*2. Where and how did the authors identify C1 and C3 chromophores in the study? And how did they measure N-containing substances. None of these experiments appear in this study.*

C1 and C3 are tryptophan and the CDOM driven by the Maillard reaction, which does not mean that C1 and C3 are the two substances. It only means that the fluorescence characteristics of C1 and C3 are the same or similar to these two substances.

According to the EEMs characteristics of the detected chromophores compared with the previous studies, we found that the maximum excitation wavelength and maximum emission wavelength of C1 and C3 are similar to these of tryptophan and the CDOM driven by the Maillard reaction, so we state that C1 and C3 chromophores are tryptophan and the CDOM driven by the Maillard reaction, respectively. Both of these CDOM are nitrogen-containing substances.

*3. Line 472-473: this sentence is confusing. The hypothesis should be reiterated here and the results stated with the implications of the work. The authors should be comparing their work with previous work on 3CDOM\* in the atmosphere in this section.*

We have corrected "*However, the above do not mean that these substances do not have the ability to form $^3CDOM^*$. In this case, as shown in Scheme 1, $^3CDOM^*$ through self-quenching and energy transfer does not consume TMP, and low-energy $^3CDOM^*$ cannot react with TMP*" to "*because self-quenching, energy transfer (as shown in Scheme 1) and low-energy do not consume TMP*".

➢ Appiani, E., Ossola, R., Latch, D. E., Erickson, P. R. and McNeill, K.: Aqueous singlet oxygen reaction kinetics of furfuryl alcohol: effect of temper- ature, pH, and salt content, Environ. Sci. Process. Impacts, 19(4), 507–516, doi:10.1039/C6EM00646A, 2017.

➢ Miljevic, B., Hedayat, F., Stevanovic, S., Fairfull-Smith, K. E., Bottle, S. E. and Ris- tovski, Z. D.: To Sonicate or Not to Sonicate PM Filters: Reactive Oxygen Species Generation Upon Ultrasonic Irradiation, Aerosol Sci. Technol., 48(12), 1276–1284, doi:10.1080/02786826.2014.981330, 2014.

---

## Author Comment (AC3) · 18 Mar 2020

Thank you very much for the reviewer's suggestions for revision. According to the reviewer's suggestions, we have revised the paper. The reviewer's comments are in blue, the answers and revised text are in black.

*Anonymous Referee #2*

*1. General comments*

*In this manuscript, the authors describe the formation of triplet state chromophoric dissolved organic matter (3CDOM\*) from a variety of aerosol samples, including laboratory generated primary organic aerosol and secondary organic aerosol, and ambient particulate matter, under simulated sunlight. Using trimethylphenol as a probe for 3CDOM\*, all aerosol samples investigated formed 3CDOM\* to a greater extent than the experimental control. The rate of formation of 3CDOM\* in the aerosol samples was correlated with chemical and optical properties of the sample including fluorescence and UV-visible absorbance. The contribution of the 3CDOM\* to form reactive oxygen species such as singlet molecular oxygen and hydroxyl radical was also quantified. The experiments described in this manuscript were thoughtfully designed and executed rigorously. However, the communication of these results should be improved before publication, as indicated in my comments below. The writing style used in this manuscript makes it challenging to read at times. For example, the authors often state a conclusion before giving context or discussing the data. This requires the reader to read through the paragraph more than once to be sure of the meaning. In addition, there are grammatical issues such as in the materials and methods section which switches between past and present tense. Better clarity in the writing would greatly improve this manuscript and better communicate these interesting results. Furthermore, some of the data was presented without sufficient discussion of its meaning and its larger scientific context. For example, discussion is lacking about why the correlation of AAE and kTMP differs between laboratory generated aerosol and ambient aerosol samples; why there is a lack of linear correlation between kTMP and contribution of C1 for the SOA samples; and the meaning of the data shown in Figures 5 and 6.*

Thanks for the reviewer's suggestion. According to your opinion we have modified the paper. It will be helpful for improving this article. It mainly includes the following aspects.

1. A larger AAE value indicates a higher polarity of CDOM and degree of oxidation;

We have added "*As shown in Figure 3, the AAE of POA and SOA are greater than Ambient PM, which indicate that CDOM of POA and SOA have a higher polarity*" in the paper.

We have corrected "Figure 2"and "Figure 3".

[Figure]

*Figure 2. The characteristics of the AAE and $E_2/E_3$ ratio of different types of aerosols.*

[Figure]

*Figure 3. The characteristics of the correlation between $k_{TMP}$, the AAE, and the $E_2/E_3$ ratio. (i), (ii) and (iii) show the correlations between $k_{TMP}$, MAE and $E_2/E_3$. (iv), (v) and (vi) show the correlations between $k_{TMP}$ and the AAE. (A), (B) and (C) are POA, SOA and ambient PM sample results, respectively.*

2. We have corrected the tense in Section;

For example, We have corrected "*Triplet state formation experiments are carried out .......*" to "*Triplet state formation experiments were carried out .......*"; We have

corrected "*……in the reactor are approximately 25 ℃ and 50%*" to "*……in the reactor were approximately 25 ℃ and 50%*".

3. $k_{TMP}$ does not varies with the content of C1 chromophore in SOA, so SOA is excluded from linear fitting.

For example, We have added "*In contrast, the proportion of the C1 component in SOA does not have a significant effect on the $k_{TMP}$, which illustrate that tryptophan do not have an effect on the formation rate of triplet state and CDOM drived by Maillard reaction may be the main substance that determines the rate of triplet state formation rate in SOA*" in the paper.

*2. Specific comments:*
*1. Lines 10–13: It is not clear what the two types of identified CDOM were and how that relates to nitrogen-containing chromophores.*

According to the comment we have corrected "*The structural-activity relationship between the CDOM type and the $^3CDOM*$ formation capacity shows that the two types of $^3CDOM*$ identified, which similar to the nitrogen-containing chromophores contributed 88% to the formation of $^3CDOM*$*" to "*The structural-activity relationship between the CDOM type and the $^3CDOM*$ formation capacity shows that the CDOM tryptophan and the CDOM driven by the Maillard reaction, which similar to the nitrogen-containing chromophores, contributed 88% to the formation of $^3CDOM*$*" in the improved paper.

*2. Line 50: The topic of the reference "You et al. 2012" does not match with the text.*

We have deleted "*$^3CDOM*$ plays an important role in the oxidation of aniline- and sulfur-containing heterocyclic pollutants (You et al., 2012)*".

*3. Lines 78–80: This section of text is very repetitive and uses vague terms "types and compositions" multiple times. Please clarify what you mean by CDOM "types and compositions".*

We have corrected "*As a precursor of $^3CDOM*$, CDOM has complex types and compositions. The types and abilities of CDOM to form$^3CDOM*$may be different, which requires us to analyze both the types and compositions of CDOM*" to "*CDOM as a precursor of $^3CDOM*$, the abilities of the different types of CDOM to form*

*3CDOM\* may be different, which requires us to analyze both the chemical composition characteristics of the different types of CDOM*".

*4. Lines 140–147: How long are the VOCs exposed to the low-oxidation and moderate oxidation conditions? The reaction time is only mentioned for the high-oxidation condition.*

The reaction time of the LO, MO and HO are 6 h. We have corrected "*The reaction time is not shorter than 6 h to ensure the complete reaction of VOCs*" to "*The reaction time of LO, MO and HO is not less than 6 h to ensure the complete reaction of VOCs*".

*5. Lines 203–204: State what you used as the "high-concentration" and "low-concentration" for TMP.*

The high-concentration of TMP is 400 μM and the low-concentration of TMP is 10 μM. We have added a state to the text, "*we compared high-concentration TMP (400 μM) with low-concentration TMP (10 μM)*".

*6. Lines 205–207: The reaction rate constant should not be expected to change with TMP concentration, but the rate of the reaction will change. The calculated k value on Figure S3 shows that the rate constant does remain essentially the same between the different concentration conditions.*

This is due to the influence of the background sample at the low concentration of TMP. Compared with the reaction system of low concentration TMP, the difference of the consumption of high concentration TMP between aerosol samples and background are more significantly different.

We have corrected "*the high-concentration TMP may have a relatively low background and a higher reaction rate constant*" to "*the high-concentration TMP may have a relatively low background and a higher reaction rate (The results are shown in Figure S3 of SI). Therefore, under the reaction condition of high concentration TMP, the aerosol sample is more significantly different from the background*".

*7. Figure 1: Legend for A (ii) does not match terminology in figure caption of "LO, MO, and HO". And the caption for graph C does not explain what data is presented.*

We have corrected "*PO, MO and OO*" to "*LO, MO and HO*" in Figure.1. And we have corrected "*(C) The average consumption of TMP by aerosols does not conform to first-order kinetics*" to "*(C) The consumption characteristic of TMP by aerosols does not conform to first-order kinetics. SB is straw burning, V is vehicle exhaust*".

*8. Lines 305–307: The first two sentences of section 3.2 seem to be contradicting each other. It is stated that the 3CDOM\* formation rate is different for each aerosol source, but then the data is presented to show that the average formation rate is the same for all aerosol sources.*

We have corrected "*The $^3$CDOM\*formation ability of different sources aerosols is different*" to "*The formation ability of $^3$CDOM\* by different sources aerosols are compared*".

*9. Lines 312–314: The TMP rate constants should be state for both the straw and wood burning samples.*

We have stated the reaction rate constants of straw and wood combustion.

We have corrected "*the $k_{TMP}$ values of the straw burning samples are lower than those of the wood burning samples (0.035 min$^{-1}$)*" to "*the $k_{TMP}$ values of the straw burning samples (0.035 min$^{-1}$) are lower than those of the wood burning samples (0.048 min$^{-1}$)*".

*10. Lines 314–316: Phenolic compounds would be expected to be present in both the wood burning samples and straw burning samples. How do you explain the difference in kTMP for these two samples?*

Phenols are not likely precursors to form triplet state. As shown in Figure 5, the contribution of C1 and C3 to triplet state formation is different, which caused difference of triplet state formation between wood burning and straw burning.

*11. Lines 324–330: Its surprising to me that the 3CDOM\* does not depend strongly on the SOA precursor. Especially since these SOA materials will have different light absorbance properties. Do you have any further insight into why the 3CDOM\* formation is so similar between these samples?*

We have mentioned "$^3$CDOM* formation of C1 is promoted by the increase in the oxidation degree" and "C3 is oxidized and decomposed by the increase in the oxidation degree" in the paper. The contribution of C2 and C4 to fluorescence varies with the degree of oxidation. However, neither of these CDOM contributes significantly to the formation of triplet state. Therefore, we insist that the fluorescence characteristics of CDOM in SOA are related to the oxidation degree of the formation of SOA, but oxidation degree do not directly affect the formation of triplet states.

*12. Lines 353–355: The strength of the correlations should be state in the text. In most cases there is only a moderate correlation between MAE or E2/E3 and kTMP.*

We expect that the relationship between $k_{TMP}$ and chemical characteristics can be studied by value. Although the correlation is moderate, aerosols from different sources basically show the same trend, which reflects the feasibility of this method.

*13. Lines 369–372: Why would the correlation of kTMP and AAE be opposite between POA, SOA, and ambient PM? This result should be discussed further.*

A larger AAE value indicates a higher polarity of CDOM and degree of oxidation. We expect to determine its ability to form triplet state based on the characteristics of the CDOM. We have added "*the AAE of POA and SOA are greater than Ambient PM, which indicate that CDOM of POA and SOA have a higher polarity*" in the paper.

*14. Line 377: "The 3CDOM* formation ability depends on the CDOM type." The data to support this claim is not presented until the following paragraph and it does not fit the topic of the paragraph it is in.*

We have deleted "*The $^3$CDOM* formation ability depends on the CDOM type*" in the paper.

*15. Line 382: The comparison of the excitation-emission peaks to tryptophan and products from the Maillard reaction should have a reference.*

We have added the references in the text.

➢ Hawkins, L. N., Lemire, A. N., Galloway, M. M., Corrigan A, L., Turley, J. J., Espelien, B. M., De Haan, D. O. Maillard Chemistry in Clouds and Aqueous Aerosol As a Source of

Atmospheric Humic-Like Substances, Environ. Sci. Technol., 50, 7443-7452, http://doi.org/10.1021/acs.est.6b00909, 2016.

➢ De Haan, D. O., Hawkins, L. N., Kononenko, J. A., Turley, J. J., Corrigan, A. L., Tolbert, M. A., Jimenez, J. L. Formation of Nitrogen-Containing Oligomers by Methylglyoxal and Amines in Simulated Evaporating Cloud Droplets, Environ. Sci. Technol., 45, 984-991, http://doi.org/10.1021/es102933x, 2010.

*16. Lines 383–384: The content of C3 in the samples is quite variable and seems to be a significant fraction in the moderate-oxidation and low-oxidation SOA samples. Could this tell you more about the SOA composition under different oxidation conditions?*

We have discussed in paper as following.

"*With increasing oxidation degree, the content of C3 decreases. The contribution of C3 to fluorescence 2.0%, 19.9% and 33.8%, which indicates that C3 is oxidized and decomposed (Wong et al., 2015). The contributions of C3 to $^3CDOM*$ are 66.4%, 33.2% and 4.8%, respectively*".

*17. Figures S7–S10 should be referenced directly in the main text.*

Figure S7-S10 have been referenced in the text.

Line 390: "*5 types of CDOM were identified through the PARAFAC model (Model error comparisons shown in Figure S7 and S8 of SI)*".

Line 409: "*The structure-activity relationship between the CDOM type and $^3CDOM*$ formation rate was established by the improved PARAFAC model in equation (2) (Residual of model as shown in Figure S9 and S10 of SI)*".

*18. Lines 417–418: It would be helpful to refer back to the reaction pathways in Scheme 1 during this discussion.*

We have corrected it in the improved paper.

We have corrected "*The quenching mechanism is mainly energy transfer, ⌞SEP⌟which means that this $^3CDOM*$ has more significant effect of driving ROS*" to "*The quenching mechanism is mainly energy transfer (as shown in (3) of Scheme 1), which means that this $^3CDOM*$ has more significant effect of driving ROS*".

*19. Lines 420–422: The lack of correlation of C1 in the SOA samples with $k_{TMP}$ should be discussed as well.*

We have added "*In contrast, the proportion of the C1 component in SOA does not have a significant effect on the $k_{TMP}$, which illustrate that tryptophan do not have an effect on the formation rate of triplet state and CDOM drived by Maillard reaction may be the main substance that determines the rate of triplet state formation rate in SOA*" in the improved paper.

*20. Line 430: It is not clear what is meant by "external mixing state of photochemical aging level".*

This statement indicates that atmospheric particles come from different sources, and the photochemical activity of particles from different sources is different. Each particle in the atmosphere may have a different photochemical activity due to different sources.

*21. Lines 431–432: The relative contribution of 3CDOM\* to overall oxidation is not shown in Table S1. I could not find this data in the paper or the supporting information.*

The formation rate of $^3$CDOM\* in the actual atmosphere are shown in last column of Table S1.

*22. Lines 445–464: This section with the data on ROS production would be better suited in the Results & Discussion section of the paper.*

Our experiments proved the driving effect of triplet state on active oxygen, and further proved the potential effect of triplet on aerosol aging. Moreover, we only conducted 2 experiments, respectively. The current experimental results are not systematic. We explained the environmental significance of studying the triplet state through this section.

*23. Lines 446–450: The meaning of the signals and the type of sample shown in Figures 5 and 6 should be explained. As well, parts (a) and (b) should be explained in the text.*

We have added "*As shown in (a) of Figure 5 and Figure 6, there are no obvious signals of •OH and $^1O_2$ without illumination*" in the improved paper.

*Supporting information:*

*1. Table S3: categories of oxidation should match text in main paper, with "low, moderate, and high oxidation".*

According to the comment we have corrected it in the improved paper.

*2. Figure S3: The label for the concentration of TMP shows $4\times10^2$ μM, but this does not match with the main text description that the TMP concentration is 4 mM ($4\times103$ μM).*

We have corrected "*4 mM*" to "*$4\times10^2$ μM*".

*3. Figure S4: Legend has incorrect spelling of 'ammonium'.*

We have corrected it in Figure S4.

[Figure]

*Technical corrections:*

*1. Line 1: "chromophore" -> "chromophoric"*

We have corrected it in the improved paper.

*2. Line 3: "driving" -> "drive"*

We have corrected it in the improved paper.

*3. Lines 6 – 8: The wording of this sentence is unclear: "Biomass combustion has the strongest 3CDOM\* generation capacity and the weakest vehicle emission capacity." Is this trying to say that vehicle emissions have the weakest 3CDOM\* generation?*

We have corrected "*Biomass combustion has the strongest $^3$CDOM\* generation capacity and the weakest vehicle emission capacity*" to "*Biomass combustion has the strongest $^3$CDOM\* generation capacity and vehicle emission capacity is the weakest*".

*4. Line 11: "structural-activity" -> "structure-activity"*

We have corrected it in the improved paper.

*5. Line 85: "expected" -> "well-suited"*

We have corrected it in the improved paper.

*6. Line 128: "is collected" -> "was collected"*

We have corrected it in the improved paper.

*7. Line 266: "were selected" -> "was selected"*

We have corrected it in the improved paper.

*8. Line 420: "Figure 3D" -> "Figure 4D"*

We have corrected it in the improved paper.

*9. Line 428: "has" -> "have"*

We have corrected it in the improved paper.

*10. Line 646: Incorrect spelling of author's name:"Canonica"*

We have corrected it in the improved paper.